# Differential dynamics of the mammalian mRNA and protein expression response to misfolding stress

Zhe Cheng[1,†], Guoshou Teo[2,3,†], Sabrina Krueger[4], Tara M Rock[1], Hiromi WL Koh[2,3], Hyungwon Choi[2,3,‡,*] & Christine Vogel[1,‡,**]

## Abstract

The relative importance of regulation at the mRNA versus protein level is subject to ongoing debate. To address this question in a dynamic system, we mapped proteomic and transcriptomic changes in mammalian cells responding to stress induced by dithiothreitol over 30 h. Specifically, we estimated the kinetic parameters for the synthesis and degradation of RNA and proteins, and deconvoluted the response patterns into common and unique to each regulatory level using a new statistical tool. Overall, the two regulatory levels were equally important, but differed in their impact on molecule concentrations. Both mRNA and protein changes peaked between two and eight hours, but mRNA expression fold changes were much smaller than those of the proteins. mRNA concentrations shifted in a transient, pulse-like pattern and returned to values close to pre-treatment levels by the end of the experiment. In contrast, protein concentrations switched only once and established a new steady state, consistent with the dominant role of protein regulation during misfolding stress. Finally, we generated hypotheses on specific regulatory modes for some genes.

**Keywords** Central Dogma; ER stress; mammalian proteomics; mass spectrometry; PECA

**Subject Categories** Genome-Scale & Integrative Biology; Membrane & Intracellular Transport

**Mol Syst Biol. (2016) 12: 855**

See also: **Y Liu & R Aebersold** (January 2016)

## Introduction

Technological advances have enabled a new generation of gene expression analysis, providing genome-wide mRNA and protein concentration data over multiple conditions or in a time course. Integrative analyses combining these complementary technologies are particularly valuable when studying the dynamics of cellular behavior in response to a stimulus, and first tools and results have emerged (Vogel *et al*, 2011; Robles *et al*, 2014; Jovanovic *et al*, 2015). In the literature, there is a growing consensus that gene expression regulation is much more intricate than assumed for many years (Vogel & Marcotte, 2012), and the exact contributions of regulation at the RNA level, that is, transcription and RNA degradation, versus regulation at the protein level, that is, translation and protein degradation, are subject to ongoing debate. Their attributable fractions range from as much as 59% for protein-level regulation to as little as 16–44% (Vogel *et al*, 2010; Schwanhausser *et al*, 2011; Li & Biggin, 2015) in steady-state cells growing under normal conditions without perturbation. In comparison, in yeast responding to various treatments, protein and mRNA expression often disagree substantially (Berry & Gasch, 2008; Fournier *et al*, 2010; Lee *et al*, 2011; Vogel *et al*, 2011; Lackner *et al*, 2012). Interestingly, this discrepancy appears to be stronger for down-regulated than up-regulated genes, hinting at the importance of protein degradation in attenuating gene expression (Berry & Gasch, 2008; Lee *et al*, 2011).

Since post-transcriptional regulation is much more intricate in mammalian cells than in yeast, for example with respect to miRNA-based translation repression or alternative splicing, such time-resolved analyses of mRNA and protein concentrations for higher organisms are particularly in demand. A few time-resolved analyses of mammalian mRNA and corresponding protein expression changes have been reported recently, for example studies that monitor the progression of mouse liver cells through the cell cycle (Robles *et al*, 2014) and the response of dendritic cells to lipopolysaccharide (LPS) treatment (Jovanovic *et al*, 2015). Although substantial protein expression changes were observed in both studies, RNA-level regulation appeared to be stronger than that of protein-level changes, fueling the debate on the relative importance of transcription, translation, and degradation.

1 Center for Genomics and Systems Biology, New York University, New York, NY, USA
2 Saw Swee Hock School of Public Health, National University Singapore, Singapore
3 National University Health System, Singapore
4 Berlin Institute for Medical Systems Biology, Max Delbrück Center for Molecular Medicine, Berlin, Germany
*Corresponding author. Tel: +65 6601 1448; E-mail: hyung_won_choi@nuhs.edu.sg
**Corresponding author. Tel: +1 212 998 3976; E-mail: cvogel@nyu.edu
†These authors contributed equally to this work
‡These authors contributed equally to this work

To quantitate the contributions of different regulatory levels and identify genes and time points at which these significant changes occur, we recently developed a statistical framework, called protein expression control analysis (PECA). PECA dissects mRNA- and protein-level regulation in time-resolved analyses and allows for consistent comparisons of the two levels of gene expression regulation (Teo *et al*, 2014). Specifically, it computes the ratio of synthesis and degradation rates over successive time intervals from paired time-course data and transforms mRNA and protein concentrations into statistical measures of regulation, as expressed by rate ratios. The rate ratios are the ratios between synthesis and degradation rates of specific molecules. The rate ratios and their changes across time provide quantitative summaries of gene expression regulation. We can use the PECA model for mRNA expression alone to characterize RNA-level regulation, or in combination with protein data to characterize protein-level regulation.

Compared to experimental measurements of protein synthesis and degradation rates using pulsed and dynamic SILAC (Doherty *et al*, 2009; Schwanhausser *et al*, 2009), PECA has the disadvantage that it currently does not distinguish between molecular synthesis and degradation, but the advantage that it does not require metabolic labeling of proteins and can therefore be applied to systems that are not amenable to SILAC. Label-free proteomics approaches are less accurate than those using isotopic labeling and therefore cannot detect small fold changes as sensitively. However, this disadvantage is effectively compensated for by recent technological and computational advances and easier sample handling that allows for the analysis of multiple replicates (Liu *et al*, 2013; Cox *et al*, 2014; Schmidt *et al*, 2014; Tebbe *et al*, 2015).

Although a few other computational approaches can quantitate the rate parameters based on first-order differential equations (Lee *et al*, 2011; Jovanovic *et al*, 2015; Omranian *et al*, 2015), PECA is the first approach that introduced a probabilistic model for statistical inference of regulatory parameters. Unlike the other approaches, PECA's probabilistic model is formulated based on Bayesian hierarchical models and leads to comparatively stable parameter estimation. More importantly, it provides a statistical score, called change point probability score (CPS), on which one can apply a score threshold associated with a desired false discovery rate (FDR) to extract genes that are significantly regulated at one or both levels. "Significant regulation" can therefore be defined as a significant change in the rates of synthesis and degradation of a gene between consecutive time intervals. The ability to estimate FDRs provides a unified analysis framework to identify mRNA- and protein-level regulation above the noise level. Using this tool, we can dissect the contribution of regulation activities at each molecular level, resulting in a final, observed protein expression trajectory.

We applied PECA to data from mammalian cells responding to stress of the endoplasmic reticulum (ER). The ER is the major protein-folding machinery and therefore highly sensitive to reagents that challenge protein folding, such as dithiothreitol (DTT). The ER stress response plays a crucial role in numerous human diseases, for example, hypoxia, ischemia/reperfusion injury, heart disease, diabetes, and neurodegenerative diseases such as Alzheimer's and Parkinson's, in which prolonged protein misfolding is detrimental to the cell (Lindholm *et al*, 2006; Yoshida, 2007). During the early ER stress response, PERK-based phosphorylation of eukaryotic translation initiation factor eIF2α causes halt of translation (Yan *et al*, 2002).

Despite this general decrease in protein synthesis, several hundreds of mRNA species increase in translation through the presence and regulation of small upstream open reading frames in the 5′UTR (uORFs)—for example, activating response of transcription factors such as ATF4 and ATF6 (Vattem & Wek, 2004; Barbosa *et al*, 2013) and active translation of the stress-related protein GADD34 (Lee *et al*, 2009)—resulting in substantial rearrangement of the transcriptome and translatome (Ventoso *et al*, 2012). The activated transcription factors then trigger downstream events, such as the unfolded protein response (UPR), a major mechanism responsible for repair and refolding of damaged proteins (Schroder & Kaufman, 2005), entailing substantial proteomic rearrangement, independent of transcription. If repair mechanisms fail, the damaged proteins are ubiquitinated and degraded by the proteasome through an ER-associated degradation pathway (ERAD) or autophagy (Imaizumi, 2007; Vembar & Brodsky, 2008; Buchberger *et al*, 2010). Prolonged or extreme ER stress, leading to an overload of the repair and degradation machineries, triggers cellular apoptosis (Han *et al*, 2013; Sano & Reed, 2013). These pathways—ER stress response, UPR, ERAD, and apoptosis—are well organized in their progression and interaction in cells, providing an ideal system for studies of the relationship between mRNA and protein expression regulation over time.

Studying mammalian cancer cells in their response to DTT over 30 h, we detected extensive regulation at both RNA and protein levels. We find that RNA-level regulation tends to be short lived and stable enough to recover the pre-treatment equilibrium between synthesis and degradation, whereas protein-level regulation is more continuous and establishes a new balance between synthesis and degradation. We also present case studies in which we generate hypotheses on the modes of underlying regulation.

## Results

### Stress treatment triggers a variety of responses across time

To compare the contributions of the mRNA and protein expression response in a dynamic system, we designed a time-course experiment of mammalian cells being subjected to ER stress. We subjected HeLa cells to 2.5 mM DTT-induced ER stress over a 30-h period, sampling at eight time points (0, 0.5, 1, 2, 8, 16, 24, and 30 h) (Appendix Fig S1). In this setup, DTT had a half-life of ~4 h (Appendix Fig S2). We first conducted a number of assays to characterize the cellular phenotype in response to the treatment (Fig 1A, Appendix Fig S3). For example, since the time course spanned more than one cell doubling of ~24 h, we tested how the stress affected cell proliferation, as measured by changes in cell density. The cell density decreased during the first 16 h, after which it increased, suggesting that a fraction of the cell population underwent apoptosis, while surviving cells proliferated normally (Fig 1A, upper panel).

This interpretation was confirmed by assays monitoring cell cycle progression and apoptosis: While apoptosis occurred during the first two hours of the experiment, later time points showed a continued division of the majority cells (Fig 1A, middle/lower panel; Appendix Fig S3). DNA labeling coupled to flow cytometry showed that apoptosis peaked at 2 h, with ~45% of cell death. Notably, the sample preparation for the mRNA and protein analysis

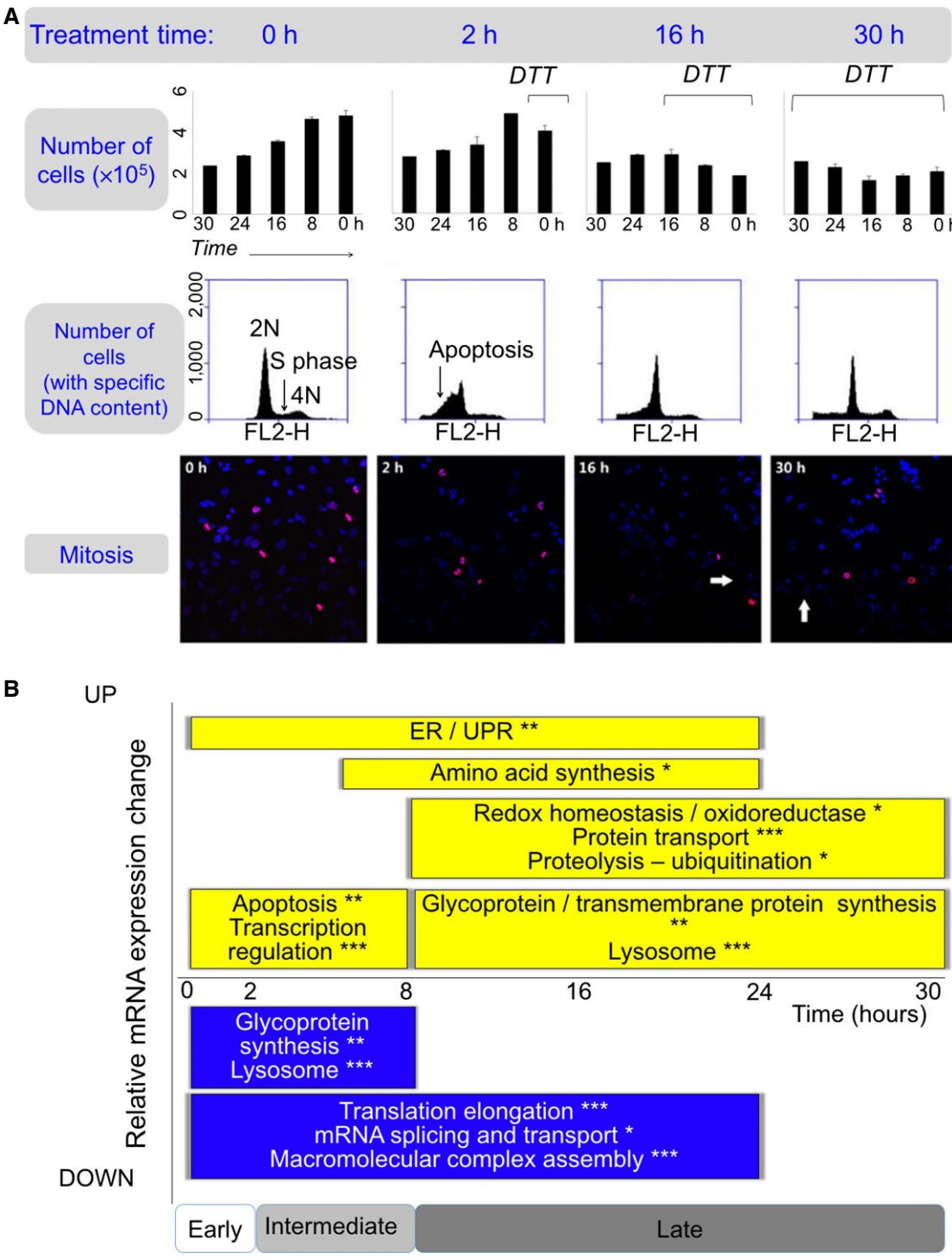

**Figure 1. Cells undergo a complex response to DTT treatment.**

While a proportion of cells were apoptotic during the first 2 h of the experiment, the majority of the cells continued cell division and displayed an extensive ER stress response.

A  We estimated the degree of active cell division based on the cell density changes, the distribution of the DNA content, and the degree of active mitosis. Top panel: Bar graphs show numbers of live cells, with mean and standard deviations. Black lines, DTT treatment time. Middle panel: Quantitative analysis of cell cycle phases by flow cytometry using propidium iodide staining of DNA for cells treated with DTT for different periods of time. The 2N, 4N peaks and S-phase plateau were observed in all time points, suggesting active cell division. Bottom panel: Immunofluorescence experiments show mitotic nuclei in red (anti-phospho-histone H3 (Ser10) antibody) and other nuclei in blue (DAPI). Mitotic nuclei were observed throughout the entire experiment. The ratio between the number of mitotic and all nuclei was similar among all the stress phases (not shown). White arrows, apoptotic nuclei. All experiments were performed in triplicate. The complete data are in Appendix Fig S3.

B  Summary of function enrichment of mRNA expression changes (FDR < 0.05, *P < 0.001, **P < 0.0001, and ***P < 0.00001). The corresponding expression data are shown in Appendix Fig S5. While some apoptosis occurred, remaining cells underwent intense unfolded protein and ER stress response.

discarded cellular debris; the results below hence focus on live cells. The same experiment also showed most of the population underwent active mitosis: As expected, most cells were in G1 stage across the entire experiment, and some cells continued DNA synthesis (Fig 1A middle panel). This result was confirmed by immunocytochemistry using the anti-phospho-histone H3 (Ser10) antibody as a mitosis marker. The stressed and control groups were very similar with respect to distribution across the G2/M checkpoint and the M phase of active cell division (Fig 1A, lower panel). In sum, while suffering from a loss of cells during the early phase of the experiment, the surviving cell population continued division throughout the entire time course.

Genome-wide transcriptomics measurements confirmed this view and manifested roughly three phases of the response where concerted changes happened: *early* (<2 h), *intermediate* (2–8 h), and *late* (> 8 h) (Fig 1B, Appendix Fig S5). Genes related to transcription regulation and programmed cell death were significantly up-regulated during the early phase (FDR < 0.05). During the intermediate phase, genes involved in ER stress and UPR were highly expressed, while at the same time, genes related to translation elongation, RNA splicing and transport, and macromolecular complex assembly were suppressed, suggesting that stressed cells put basic cellular functions to a halt (FDR < 0.05). During the late phase, cells expressed genes involved in protein ubiquitination, lysosome, and glycoprotein and transmembrane protein synthesis, indicating the recovery of surviving cells (FDR < 0.05). The increase in lysosomal proteins is consistent with the observations which found that the UPR remodels the lysosome as part of a pro-survival response (Ron & Hampton, 2004; Sriburi *et al*, 2004; Brewer *et al*, 2008; Elfrink *et al*, 2013).

**The integrated transcriptome and proteome are highly dynamic**

Next, we conducted a large-scale, quantitative proteomic analysis to complement the transcriptomic data. A variety of tests confirmed the quality of the proteomic data, for example, Western blots of selected proteins and analysis of housekeeping genes, and its reproducibility across the two biological replicates (Appendix Figs S11–S13). We quantitated a total of 3,235 proteins at least once across all time points and replicates and chose a high-confidence dataset of 1,237 proteins with complete time-series measurements across both replicates for further analysis. This high-confidence dataset is comparable in size to that of a recent study (Jovanovic *et al*, 2015). We also constructed an extended dataset with 2,131 proteins which showed similar results (Appendix Fig S19).

The high-confidence dataset was further processed to remove measurement noise and then used for the analyses described below. Protein concentrations spanned about five orders of magnitude (Appendix Table S1), which is similar to what other large-scale studies observe (Schwanhausser *et al*, 2011). Their reproducibility was high (*R* > 0.94 for seven of the eight time points, Appendix Fig S10); the correlation with the corresponding mRNA concentrations was consistent across samples (Appendix Fig S13). Heatmaps of the integrated and clustered mRNA and protein expression values show that overall expression changes were similar between the two biological replicates (Fig 2, Appendix Figs S5, S9 and S14), but some discrepancies existed. In some cases, peak expression changes occurred at 2 h in one replicate and at 8 h in the other. To describe

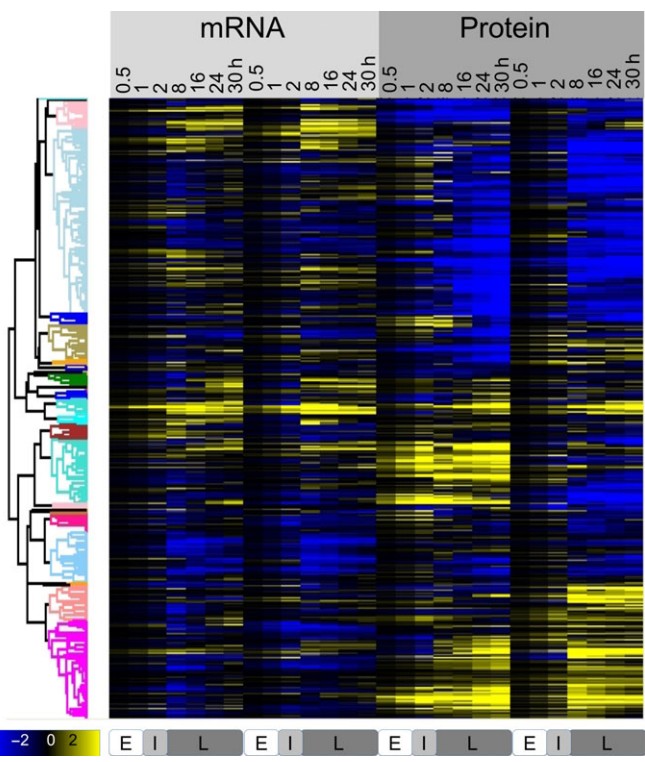

**Figure 2.  RNA and protein expression changes are highly dynamic.**
The heatmap shows the normalized, relative expression values for both mRNA and protein measured across two replicates (*N* = 1,237), log-transformed (base 10). Profiles were clustered as described in Materials and Methods; the cluster definitions are provided in Dataset EV1. Bottom labels E, I, L mark the early, intermediate, and late phase, respectively.

experimental reproducibility, we calculated a replicate consistency measure (RCM) that lists the Pearson's correlation coefficient between replicate time-series measurements of normalized, log-transformed RNA and protein concentrations. At a total of eight data points, a Pearson's correlation coefficient > 0.7 corresponds to a *P*-value = 0.05. For example, for GRP78, the RCM is 0.87/0.97, suggesting high reproducibility between the two biological replicates. Appendix Fig S13 displays the frequency distributions of all RCM values and shows a bias toward high values.

In Fig 2, we identified several major groups with similar expression changes. For example, genes involved in the general stress response were significantly up-regulated during the intermediate and late phase of the experiment both at the mRNA and at the protein level (Appendix Fig S14). Translation-related and mitochondrial genes were down-regulated at the mRNA level, consistent with a halt in metabolic processes of stressed cells; however, these proteins were up-regulated at the protein level.

**A statistical tool identifies hidden regulatory signals**

In the results described below, we used the PECA tool to extract regulatory signals from the RNA and protein time-series data. First, to illustrate the interpretation of PECA results, we show the example of GRP78 (HSP5A), an ER chaperone induced by ER stress and an important anti-apoptotic, pro-survival component of the UPR

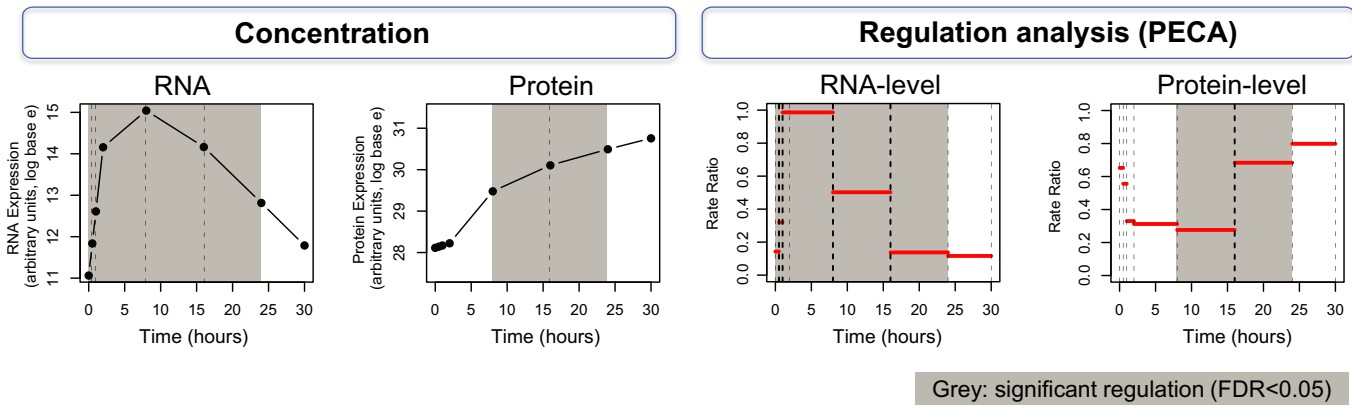

**Figure 3.  PECA deconvolutes expression data to extract regulatory information at the RNA and protein level.**
The example shows the chaperone GRP78, a key ER stress protein. mRNA and protein concentrations are shown on the left; PECA results are shown on the right for RNA and protein level, respectively. Intervals with significant regulation as determined by PECA are gray shaded (FDR < 0.05). The value of PECA is illustrated at the 16-h time point at which mRNA concentration decreases, while protein concentration still rises. PECA highlights that there is a significant RNA- and protein-level regulation around this time point—a signal that would otherwise likely have been overlooked.

(Fig 3). The figure displays GRP78's mRNA and protein concentrations and the PECA results with respect to RNA- and protein-level rate ratios and significance (RCM = 0.87/0.97). We see that GRP78's mRNA and protein expression patterns across the treatment were very different from each other: mRNA concentrations peaked at 8 h and declined afterward, while protein concentrations continuously increased. Similar to the concentration data, RNA rate ratios for GRP78 peaked between two and eight hours and decreased later, while protein rate ratios plummeted in the beginning and elevated to the pre-treatment level throughout the intermediate and late phase, resulting in continuously rising protein concentration. PECA identified both significant regulation of RNA expression in the early and late phase, respectively, as well as a significant protein-level regulation in the late phase of the experiment (FDR < 0.05; Fig 3, shaded area).

Importantly, PECA identified what was invisible from the inspection of concentration data alone: At around 16 h, RNA expression was significantly down-regulated, but protein concentrations continued to rise. This increase was realized through an up-regulation of protein expression, either through increased translation or through protein stabilization, and PECA sensitively identified this regulatory event. Notably, PECA was able to distinguish this up-regulation at the protein level from an increase in protein concentrations that is purely due to constant translation of the existing mRNAs at preceding time points, and define regulation as a significant *change* in synthesis and degradation rates from one time interval to the next. This regulatory event is also an example of the sometimes counterbalancing effects of RNA- and protein-level regulation (discussed below and in Appendix Fig S16). Incorporating overall data properties and measurement noise, PECA enabled us to quantitate regulatory events and extract them in a systematic and statistically consistent manner. The entire PECA results are provided in the Dataset EV1.

### Protein concentration changes occur in greater magnitude, but both regulatory levels contribute equally and independently

Before discussing the overall PECA outcomes, we examined general properties of the integrated mRNA and protein concentration data

(Fig 4A–D). In general, both protein and mRNA concentrations hardly changed during the early phase of the experiment, but during the intermediate and late phase with different dynamics. Consistent with earlier studies (Murray *et al*, 2004), the transcriptome was comparatively static in our experiment, with average changes of about 1.5-fold. Transcript concentrations diverged maximally from the steady state at 8 h, after which they returned to the original levels. In contrast, protein concentrations continuously diverged from the beginning until the end of the experiment, with much less change during the late phase (Fig 4, Appendix Fig S15). The magnitude of change was also more pronounced for proteins than for mRNAs, illustrated by the average (and range) of expression fold changes which were larger than those for mRNAs (Fig 4, Appendix Table S1).

To quantitate the contribution of the two regulatory levels to the cellular response in this system, we extracted significantly regulated genes by applying a 5% FDR cutoff to the PECA results. Figure 4E and F shows the number of significantly regulated genes per time point; Table 1 summarizes the results in a different manner. Most of the significant RNA-level regulation during the ER stress response occurred during the intermediate and also during the late phase (Fig 4, Table 1). Regulatory activity, that is, changing mRNA rate ratios, spiked around the 2- to 8-h mark, without additional regulation afterward: Concentrations simply returned slowly back to initial values. A similar overall pattern was also observed for the protein level (Fig 4).

Table 1 shows the numbers of significant regulatory events for one of the replicates, grouped according to phase, level, and direction of the regulation. While most changes occurred during the intermediate phase, the distributions of these changes are consistent across phases and replicates even when different significance cutoffs were applied (not shown). The numbers are symmetrically distributed across the table, confirming the observation from Fig 4E and F that mRNA- and protein-level regulation contributes equally to the overall gene expression changes in this experiment, affecting similar numbers of genes. As Table 1 shows, if a gene was significantly regulated during a specific phase of the response, this regulation typically occurred at either the mRNA or the protein level, but not

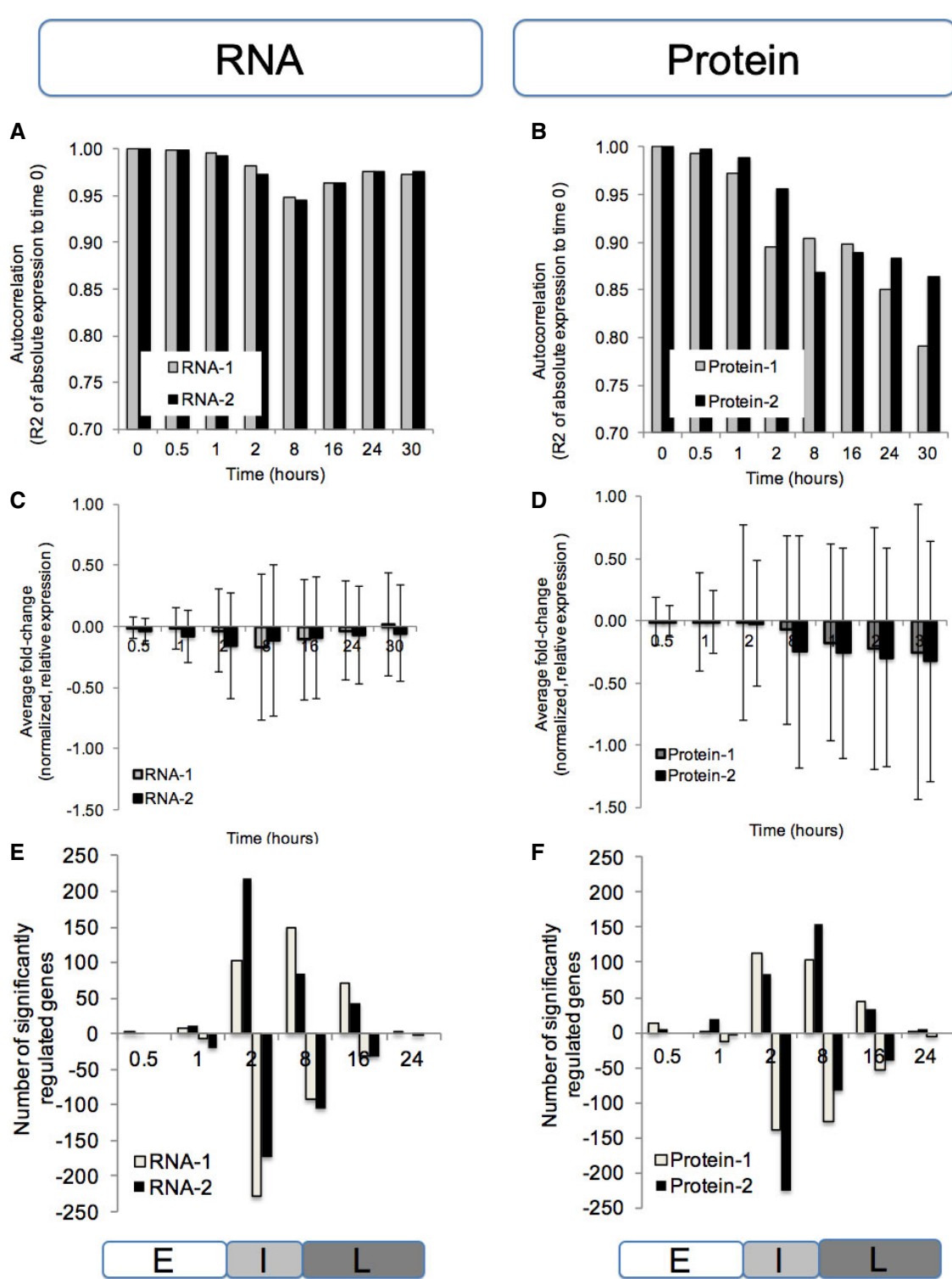

**Figure 4.  The proteome response is dominant during ER stress.**

The concentrations diverge more strongly in the protein data compared to the mRNA data with respect to magnitude (A–D), but both mRNA and protein show similar numbers of significantly regulated genes (E, F).

A, B   Correlation (Pearson's $R^2$) between normalized, absolute expression values at time 0 and the respective time points.

C, D   Average fold change (log base 10) and standard deviation of normalized, relative expression values.

E, F   The number of significantly regulated genes as determined by PECA (FDR < 0.05). We summarized the CPS probabilities of each gene by choosing the maximum probability across the time points in each of the three phases, which allows us to characterize how expression regulation (rate ratio) has shifted phase by phase. Labels E, I, and L mark the early, intermediate, and late phase, respectively.

                                      

**Table 1. RNA- and protein-level regulations contribute equally to gene expression.**

| | Early | | | Intermediate | | | Late | | | Scale |
|---|---|---|---|---|---|---|---|---|---|---|
| Protein | | | | | | | | | | >500 |
| RNA | Down | None | Up | Down | None | Up | Down | None | Up | >200 |
| Down | 0 | 7 | 0 | 26 | 180 | 21 | 35 | 63 | 16 | >50 |
| None | 13 | 1194 | 15 | 99 | 726 | 83 | 116 | 728 | 94 | >20 |
| Up | 0 | 8 | 0 | 15 | 77 | 10 | 23 | 138 | 24 | >0 |

Using PECA, we extracted genes that are significantly regulated at the RNA level, the protein level, at both levels, or neither (FDR < 0.05). The tables group these genes into the three different phases ("early", "intermediate", and "late") and distinguish between up- and down-regulation, marked by "Up" and "Down", respectively. Most changes occur during the intermediate phase. The distribution of the numbers across the tables is symmetric, indicating that mRNA- and protein-level regulations are equally important.

both at the same time; the numbers of genes in each of the square's corners are smaller than those in the middle rows or columns. However, some genes showed mRNA- and protein-level regulation moving in the same direction during the same phase, and others showed movement in opposite directions.

Table 1 already indicates that discordant regulation is comparatively rare: Only few genes are listed in the lower left and upper right corners of the tables (75 genes in total). One such example is GRP78 (Fig 3) for which mRNA expression is down-regulated and protein expression is up-regulated at the 16-h time point. An alternative way to identify discordant regulation confirmed this result, that is, via filtering for negative correlation between PECA's mRNA and protein time-course rate ratios in both replicates (Dataset EV1, Appendix Fig S16A and B). We then further refined this filtering and required not only opposing regulation, that is, at least one significant regulatory event at the mRNA and one at the protein level, but also constant protein concentrations, that is, changes smaller than 1.5-fold across both biological replicates. Such a scenario would indicate cases of "buffering" in which changes in mRNA concentrations are counterbalanced to result in no overall change at the protein level. Three out of the 75 genes passed this additional filtering and are shown in Appendix Fig S16C. One of these genes is HSC70 (RCM = 0.91/0.09), a chaperone discussed below (Fig 6A). Overall, we conclude that discordant regulation is rare, and the dynamics in the balance of synthesis and degradation of mRNA and protein occur in an independent manner.

**Protein expression regulation reaches a new steady state**

After quantitating the overall contributions and direction of the regulatory changes, we set out to examine general temporal patterns of regulation. To do so, we constructed a clustered heatmap of median-centered RNA and protein rate ratios and calculated the average rate ratios across the six largest clusters (Fig 5). A stark contrast in coloring between consecutive columns indicates significant regulation of an individual gene: A change in synthesis and degradation rates results in a change in rate ratios between time intervals. Fig 5 shows a striking difference between the mRNA and protein level of regulation. For RNA-level regulation,

many PECA rate ratios spike during the intermediate phase, resulting in significant changes at both the two- and eight-hour boundary time points. Before and after this interval, mRNA synthesis and degradation rates were relatively constant, with some exceptions during the late phase. We note that absence of regulation in the early time points is unexpected since, for example, many cells underwent apoptosis within the first two hours, suggesting that these processes may have occurred before our first measurement at 30 min. The pulse-like or transient behavior of the RNA-level regulation was confirmed both for the extended dataset (2,131 genes) and for the entire transcriptome (> 18,000 genes) (Appendix Figs S19 and S21), indicating that the high-confidence dataset delivers representative results. We observe strong spikes in extreme rate ratios between 2 and 8 h, with significant regulation leading into and out of this phase.

Next, we analyzed the temporal behavior of protein-level regulation during our experiment. Similar to mRNA, little regulation occurred during the early phase, but it rapidly increased during the intermediate phase (Fig 5). However, in contrast to the pulse-like mode of RNA-level regulation, PECA showed that many protein rate ratios changed only once during the intermediate phase, in a switch-like or permanent manner, but then remained constant. This switch-like behavior is even more apparent when examining the averaged rate ratio changes across the different gene expression clusters (Fig 5, right). After the change at around 2 h, the protein concentrations did not revert back and stayed at the new level throughout the remainder of the experiment, indicating execution of the same protein synthesis and degradation rates that had been set earlier, without additional regulation. As can be seen in Fig 5 (right), the switch-like behavior applied to both up- and down-regulation and was independent of the mode of mRNA regulation. It is also present in the extended dataset (Appendix Fig S19). The PECA results confirmed what the concentration data had hinted for: While mRNA expression returned to the original values, protein-level regulation reached a new steady state.

**PECA results help to generate hypotheses on regulatory modes**

Finally, we examined three groups of genes in detail to illustrate how our analysis can detect signals that are otherwise hidden and help to generate hypotheses on possible regulatory modes. The first example group includes GRP78 (HSPA5, BiP; RCM = 0.87/0.97) and other chaperones (Fig 6A). As discussed above, up-regulation of GRP78 at both the mRNA and protein level is expected due to its crucial role during the ER stress response. It is tempting to hypothesize that its strong protein-level up-regulation might be mediated by the internal ribosome entry site in its 5′UTR. However, the validity of this hypothesis is still debated (Fernandez *et al*, 2002).

Another gene in this group is HSC70 (HSPA8; RCM = 0.91/0.09), which is, similar to GRP78, a chaperone with pro-survival functions in the cell (Zhang *et al*, 2013). However, its protein expression pattern is different from that of GRP78 in that it remains constant across the time course. HSC70 is constitutively expressed and helps folding of nascent protein chains. Under stress, it has been described to be slightly induced (Liu *et al*, 2012). In our dataset, we observe a significant drop in mRNA concentrations during the early phase of the experiment and a later recovery. Interestingly, this expression

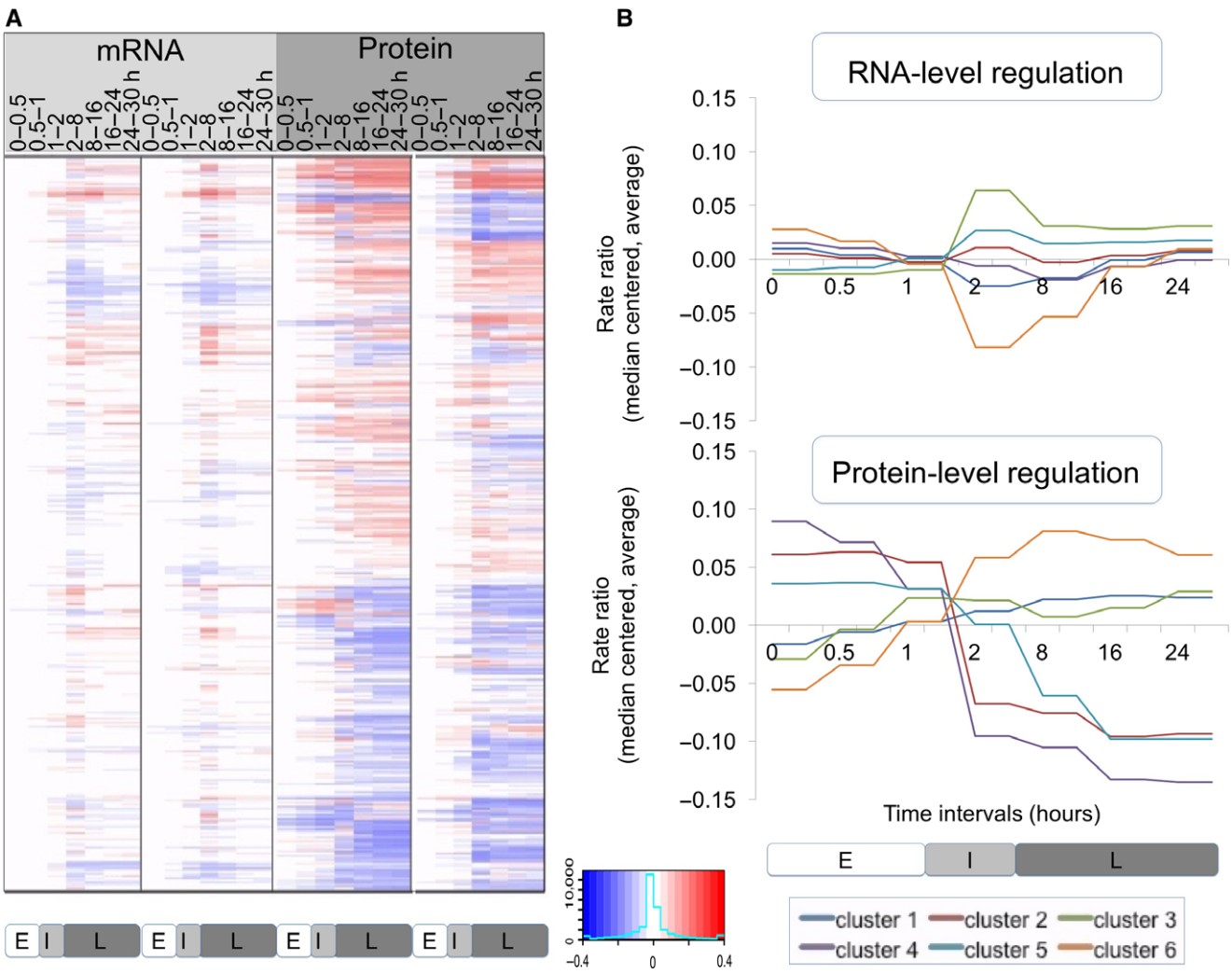

**Figure 5.  RNA- and protein-level regulations have different temporal modes.**

The predominant regulatory level of protein synthesis and degradation shows a switch-like behavior that leads to a new steady state.

A   Heatmap of RNA and protein rate ratios as computed by PECA, shown for the two replicates.

B   The average rate ratios across six major clusters for both RNA (top) and protein (bottom). RNA rate ratios show a spike in their changes during the intermediate phase, while protein rate ratios change only once around the two-hour mark and remain at the new steady-state level throughout the remainder of the experiment. The clusters are defined in Dataset EV1.

change is not transmitted to protein concentrations, but counterbalanced by a significant, transient up-regulation of protein expression. This behavior makes HSC70 one of the three examples for potential buffering discussed above (Appendix Fig S16).

Not only HSC70, but also HSP90AA1 and HSP90B1 serve as co-chaperones for the HSP90 proteins. HSP90B1 (GRP94, TRA1; RCM = 0.90/0.91) is localized to melanosomes and the ER and assists in protein folding. The protein appears to be regulated in two phases. After a short-term transcription increase (followed by transcription decline), protein production is augmented during the intermediate and late phases of the ER stress experiment. Finally, Fig 6A shows P58IPK (DNAJC3; RCM = 0.88/0.63), which is a member of the Hsp40 chaperone family and an inhibitor of the eIF2α kinase PERK. Due to this function, it is essential for translation re-start after the initial, ER stress-related translation shutdown (Roobol *et al*,

2015). An ER stress element in P58IPK's promotor region is known to activate the gene's transcription in response to ER stress (Yan *et al*, 2002). In our experiment, despite up-regulation at the mRNA level, protein concentrations are constant over the entire time course, suggesting homeostatic down-regulation at the protein level. However, this case did not qualify for buffering according to our criteria. The low P58IPK levels together with the continuous increase in GRP78 concentration (Yan *et al*, 2002) indicate that an ongoing ER stress response delayed return to normal translation in our experiment.

The second example group comprises 196 genes with invariable RNA concentrations, but whose protein concentrations increased during the late phase (Appendix Fig S14, Dataset EV1, cluster 8). Genes in this group are enriched in mitochondrial proteins, ATP biosynthesis, ribosomes, translation, and transmembrane

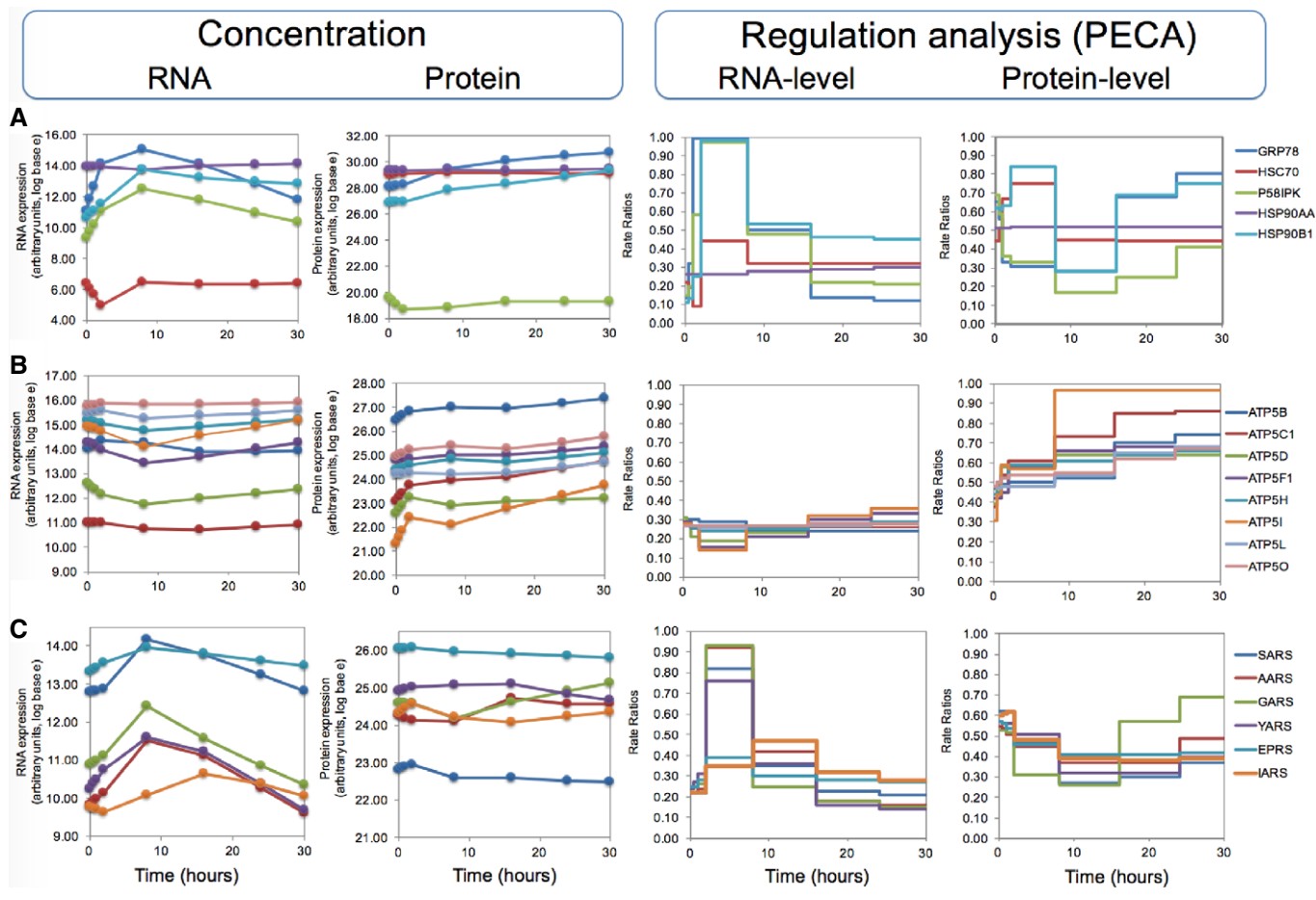

**Figure 6. PECA identifies groups of similarly regulated genes.**
mRNA and protein concentrations are shown on the left; PECA results are shown on the right for RNA and protein level, respectively.

A Five chaperones, including GRP78, with mixed expression patterns.
B Eight subunits of ATP synthases observed in the experiment with mostly invariable RNA concentrations and increasing protein concentrations. PECA amplifies the hidden signal and identifies a significant protein-level regulation.
C Six aminoacyl-tRNA synthetases whose mRNA concentration increases temporarily, but the protein concentrations remain largely constant. PECA deconvolutes the two opposing regulatory effects that act at the RNA and protein levels.

proteins (FDR < 0.05). The ATP synthase genes are shown in Fig 6B. ATP synthases have essential roles in cellular ATP biosynthesis, and their increased activity likely boosts cellular ATP levels, which in turn helps provide the energy needed for the UPR. We identified eight subunits (ATP5B, C1, D, F1, H, I, L, O; average RCM = 0.50/0.61) with similar expression patterns. PECA of these genes shows how our tool can extract an otherwise hidden signal: PECA correctly identified a significant positive regulation at the protein level that results in an increase in absolute protein concentrations of the ATP synthase subunits.

To generate hypotheses on possible mechanisms for the up-regulation of these proteins, we collected > 160 sequence features, including length, signal sequences, nucleotide composition, amino acid composition, translation regulatory elements, RNA secondary structures, and post-translation modifications (Appendix Table S2). When testing this example group for biases across the features, we found a significant depletion in proline and glutamic acid, which are parts of PEST sequences that shorten protein half-lives, and disordered regions, that is, COILS and REM465 (*t*-test, *P* < 0.0001),

which are also known to destabilize proteins. Depletion in these two characteristics would stabilize the protein and would explain the up-regulated protein expression found by PECA.

The last example group contains 91 genes (Dataset EV1, cluster 3) that are characterized by an increase in both mRNA and protein concentrations and are significantly enriched in oxidoreductases and interestingly, aminoacyl-tRNA synthetases, namely GARS, YARS, IARS, AARS, SARS, and EPRS (FDR < 0.05; average RCM = 0.88/0.21). The aminoacyl-tRNA synthetases are shown in Fig 6C and are examined in more detail in Appendix Figs S17 and S18. A number of the enzymes show a striking gene expression pattern in which protein synthesis is delayed by several hours, compared to RNA synthesis. As this protein synthesis only occurs after mRNA concentrations decrease already, the resulting final protein concentrations remain comparatively constant (Fig 6C). These cases did not qualify for "buffering" regulation, as they did not pass our filtering criteria.

However, post-transcriptional regulation of aminoacyl-tRNA synthetases has been observed before in other contexts (Kwon *et al*,

2011; Chen *et al*, 2012; Park *et al*, 2012; Guan *et al*, 2014; Wei *et al*, 2014). Its cellular role and underlying mechanism remained unknown until a recent publication delivered an intriguing explanation: Aminoacyl-tRNA synthetases express alternative splice variants that lack the catalytic domain but which often have additional "moonlighting" functions independent of their original role during translation (Lo *et al*, 2014). Based on these findings, we hypothesized that the discrepancy between mRNA and protein expression patterns for some genes might be explained by the differential expression of splice variants, and we examined the proteomics data manually for such examples (Appendix Figs S17 and S18). Unfortunately, as the proteomics experiment had not been designed to detect splice variants, only three enzymes (AARS, IARS, and QARS) provided enough information to draw some conclusions. While we detected for each of these three enzymes a set of sequence variants with differential expression, future work will have to confirm whether these alternative splicing events are indeed functional and affect the overall, averaged protein expression levels as observed in Fig 6C.

## Discussion

After much debate on the relative contributions of RNA- and protein-level regulation to set steady-state protein concentrations (Vogel *et al*, 2010; Vogel & Marcotte, 2012; Li *et al*, 2014; Csardi *et al*, 2015; Jovanovic *et al*, 2015), it is time to start examining a new dimension: that of time-resolved expression changes. However, such datasets are still rare, in particular for mammalian cells. Using quantitative proteomic and transcriptomic data of mammalian cells responding to DTT and a statistical tool specifically designed to analyze time-series protein and RNA measurements, we deconvolute the relative contributions of transcription, translation, and molecule degradation to changes in expression during a 30-h time-course experiment. Our analysis focuses on changes after the first 30 min of the response; regulation before the first half-hour time point has also been described (Satpute-Krishnan *et al*, 2014). Further, we focus on results that are consistent across the two biological replicates and that are observed in a high-confidence dataset without missing data. We have used our statistical tool, PECA, to define expression regulation in a quantitative manner, extracting significantly regulated genes and their corresponding time points. While the major results from this study have been confirmed by analysis of the total transcriptome (Appendix Fig S21) and an extended mRNA/protein dataset (Appendix Fig S19), our discussions are still restricted to a comparatively small subset of the proteome. However, the use of complete time-series data and biological replicates increases our confidence in the validity and generality of our findings.

Overall, the transcriptome in our dataset was comparatively static, consistent with earlier observations (Murray *et al*, 2004). In contrast, as expected from a treatment that affects the protein homeostasis function of the ER, we found that protein concentrations changed more drastically than those of mRNA. However, despite the smaller RNA concentration changes, we found that mRNA- and protein-level regulation contributed equally to the final expression response (Table 1). This finding contrasts the reports from steady-state systems in yeast (Li *et al*, 2014; Csardi *et al*, 2015), but is consistent with a recent study on a dynamic system of mammalian cells responding to lipopolysaccharide (LPS) treatment (Jovanovic *et al*, 2015). In both the ER stress response described here and the published data on the LPS response, regulation at both RNA and protein level contributes to the change in the system, and protein expression changes drive the synthesis and turnover of highly abundant molecules (Appendix Figs S22 and S23, Appendix Tables S1, S3 and S4).

While most concentration changes were seemingly concordant between mRNA and protein in terms of the outcome at the end of the experiment, many regulatory (rate ratio) changes, in particular the most pronounced ones, were independent between the mRNA and protein level with respect to their timing or direction. We even observed a small number of cases in which transcript- and protein-level regulation acted in opposite directions. One example is the ER stress-related chaperone GRP78, whose mRNA concentration was in decline at 16 h, while protein concentrations still increased (Fig 3). We find that discordance at a specific time point was often resolved by a simple delay in the response: The changes at the RNA level are initially counteracted at the protein level, but later supported by concordant action. Overall, true "buffering" appears to be the exception within the set of proteins we surveyed, and most regulatory events at the mRNA and protein levels are coordinated, albeit with different timing, to achieve a new proteomics state.

Most strikingly, we found that the mRNA- and protein-level regulation during the ER stress response, while equal with respect to the number of significant genes (FDR < 0.05), presented itself via different temporal patterns. mRNA concentrations responded in a "pulse-like" fashion, transiently coordinating changes in RNA concentrations which returned to original levels by the end of the 30-h measurement. In comparison, protein regulation altered in a "switch-like" manner, permanently changing to a new steady state that was different from the original state. We note that without higher temporal resolution, it is impossible to know if such a switch is indeed very rapid or more continuous over a period of time.

We aimed to estimate the generality of these findings by re-analyzing the published dataset on the LPS response with PECA (Appendix Figs S22 and S23, Appendix Tables S3 and S4) (Jovanovic *et al*, 2015). This re-analysis confirmed that the LPS response is driven by substantial RNA regulation immediately after stimulation (Jovanovic *et al*, 2015). However, we also found that the changes in protein concentrations were not entirely accounted for by RNA regulation alone: The rates of translation and protein degradation also changed significantly and fine-turned the final protein concentrations. Again, we observed a pulse- and switch-like behavior similar to that of the ER stress response, suggesting transient and permanent regulatory changes, respectively. However, in contrast to the ER stress response in which proteins appeared to switch from the original to a new steady state, in the LPS response we found the switch-like behavior for RNA-level regulation. Given that the LPS response is driven largely by transcription, while the ER stress response strongly affects the proteome, we hypothesize that the switch-like regulatory patterns occur in the dominant level of regulation to foster major and perhaps permanent changes in the cellular state. Future work will have to test the validity of this hypothesis.

Mapping a set of sequence and experimental datasets to groups of proteins with similar expression patterns, we generated hypotheses on the differential regulation of these genes. Interestingly, for a

set of genes with a significant up-regulation at the protein level without matching changes at the mRNA level (e.g., example group two, Fig 6B), we found depletion of destabilizing signatures such as PEST signals or disordered regions. This depletion suggests that sequence evolution and the dynamic response to a stimulus operate cooperatively, enabling the accumulation of proteins under stress.

This example group was also enriched in specific protein functions, for example, mitochondrial ATP synthases, which offer another intriguing interpretation for their differential regulation. Mitochondrial proteins, such as ATP synthases, are thought to be preferentially translated near the organelle (Margeot *et al*, 2005; Smits *et al*, 2010; Rak *et al*, 2011) and indeed, when examining published data on localized translation in yeast, we find that many yeast orthologs of the ATP synthase complex are translated near the mitochondria (Williams *et al*, 2014) (not shown). This localized translation differs from cytoplasmic translation and occurs in several biological processes, for example, ejaculated sperm (O'Brien, 2003; Gur & Breitbart, 2008). RNAs encoding mitochondrial proteins account for almost 30% of localized translation and produce proteins with functions in the mitochondria (Deglincerti & Jaffrey, 2012). We hypothesize that the mitochondrial vicinity may counterbalance the redox imbalance caused by the DTT treatment in our experiment and enable translation, while protein synthesis is repressed in the remainder of the cytosol (Wallace *et al*, 2010; Liang *et al*, 2013; Venditti *et al*, 2013). Continued expression of ATP synthase genes in turn would enable the cells to provide the vast amounts of energy needed to sustain the stress response.

In sum, both technology and statistical tools to analyze time-series transcriptomics and proteomics measurements have advanced enough to allow for first insights into the dynamics of gene expression regulation. In this work, we have demonstrated that protein misfolding stress places much greater weights on the importance of protein-level regulation than the recent observations that characterized transcriptomics changes as the main driver of phenotypic adaptation (Lee *et al*, 2011; Jovanovic *et al*, 2015). We also demonstrated that in our experiment, mRNAs and proteins are regulated with different temporal patterns, with the dominant protein response adhering to a switch-like behavior that establishes a new steady state in the cell. We therefore suggest that the way RNA- and protein-level regulation determines the post-treatment homeostatic condition depends on the nature of treatment and its implications on the fitness of the cells, that is, stress conditions versus stimulation. The debate on the relative contribution of transcriptomics and post-transcriptional regulation will have to be continued in a context-specific discourse with systematic comparisons of various conditions.

# Materials and Methods

### Cell culture and experimental setup

HeLa cells were cultured in DMEM (Sigma) with 10% fetal bovine serum (Atlanta Biologicals) and 1× penicillin–streptomycin solution (Corning Cellgro) at 37°C and 5% $CO_2$. At approximately 60% confluency, dithiothreitol (DTT) at 2.5 mM concentration was added to induce stress for different periods of time, that is, 0, 0.5, 1, 2, 8, 16, 24, and 30 h. To account for different cellular ages at harvest, the

experiment was conducted so that the experiment started with the 30-h treatment period, and the cells were then collected at the same time (Appendix Fig S1). This protocol ensured that all cells were cultured for the same time period.

### Cell counting

The cells were seeded in parallel plates 3 days before sample preparation and trypsin-digested from the plates using 0.5% trypsin for 2 min. Trypan blue (GIBCO, Life Technologies, USA) was used to label living cells and the cells were counted using a hemocytometer. The cells were treated with 2.5 mM DTT at designed time points, collected, and counted. The assay was conducted in triplicate, and the average and standard deviation were calculated for final results.

### DNA staining and flow cytometry

The control and stressed cells were digested from plates with 0.5% trypsin and prepared as single-cell suspensions in 1× Dulbecco's phosphate-buffered saline (DPBS) solution. The cells were then fixed and permeabilized with 70% ethanol for 2 h at 4°C. For flow cytometry analysis, cells were rehydrated in DPBS and incubated with RNase A (ribonuclease A, 19101, QIAGEN) for 30 min at 37°C to digest cellular RNA and thus decrease background RNA staining. After RNase A treatment, the fluorescent molecule propidium iodide (PI) was added into the cell suspension at 50 µg/ml concentration and incubated for 30 min at room temperature to bind DNA unspecifically. Cells with stained DNA were quantitated by flow cytometry analysis on an FL2 flow cytometer with 488-nm laser excitation. The assay was conducted in duplicate.

### Immunocytochemistry

The cells were cultured on glass cover slips in cell culture dishes with conditions identical to those described above, DTT-treated, and fixed with fresh 4% paraformaldehyde for 10 min. The fixed cells were pre-incubated in 0.1 M DPBS containing 10% normal donkey serum and 0.2% Triton X-100. Condensed nuclear DNA was labeled with anti-phospho-histone H3 (Ser10) antibody (06-570, EMD Millipore, MA, USA). The primary antibody incubation was conducted at 4°C overnight in the medium containing 5% normal donkey serum, 0.2% Triton X-100, and 1% bovine serum albumin. After washing with PBS, the binding sites of the primary antibodies were revealed by incubating for 2 h at 4°C with the secondary antibody, rhodamine red-X (RRX)-conjugated donkey anti-rabbit IgG. The samples were mounted by ProLong Gold antifade reagent with DAPI (P36935, Life Technologies, OR, USA) and scanned with a Leica SP5 confocal laser scanning microscope (CLSM, Leica, Mannheim, Germany). To avoid reconstruction stacking artifacts, RRX and DAPI were evaluated by sequential scanning of single-layer optical sections.

### Transcriptomics measurements

To estimate absolute mRNA expression values, Agilent-028004 SurePrint G3 Human GE 8x60K microarrays were used. RNA extraction

was conducted using Trizol (Sigma) followed by the use of phase separation using the Phase Lock Gel Heavy (5-Prime, manufacturer's protocol). RNA was then purified using the RNA MinElute Kit (QIAGEN), and a Nanodrop ND-1000 was used to quantitate RNA. Cyanine-3 (Cy3)-labeled cRNA was prepared from 50 ng RNA using the One-Color Microarray-Based Gene Expression Analysis (Low Input Quick Amp Labeling) Protocol (Agilent) according to the manufacturer's instructions, followed by RNAeasy column purification (QIAGEN). Dye incorporation and cRNA yield were estimated with Nanodrop; 600 ng of Cy3-labeled cRNA was fragmented at 60°C for 30 min in a reaction volume of 25 μl containing 1× Agilent fragmentation buffer and 2.5× Agilent blocking agent following the manufacturer's instructions. Upon completion of the fragmentation reaction, 25 μl of 2× Agilent hybridization buffer was added to the fragmentation mixture and hybridized to Agilent Whole Human Genome Oligo Microarrays (G4112A) for 17 h at 65°C in a rotating Agilent hybridization oven. After hybridization, microarrays were washed for one minute at room temperature with GE Wash Buffer 1 (Agilent), for one minute at 37°C with GE Wash buffer 2 (Agilent), and for 10 s in acetonitrile. Slides were scanned immediately after washing using the Agilent DNA Microarray Scanner (G2565CA) with one-color scan setting for 8x60k array slides (scan area 61 × 21.6 mm, scan resolution 3 μm). The scanned images were analyzed with Feature Extraction Software 10.7.3.1 (Agilent) using default parameters (protocol GE1_107_Sep09 and Grid: 028004_D_F_20110819). To confirm the accuracy of the transcriptomics experiments, selected time points were compared to data collected from the same samples, but using RNA-seq (Appendix Fig S8).

## Transcriptomics data processing and quality control

Upon data collection, probeset identifiers were mapped to Ensembl transcript and gene identifiers, and intensity data were averaged to obtain one value per gene. The data were then log-transformed (natural log) and quantile-normalized. A jackknife procedure was devised to remove aberrant expression measurements. In the procedure, one datum was removed at a time and the total range of variation (TRV), defined as the difference between maximum and minimum, for the remaining dataset was recorded. After following this procedure for all data points, the ratio of the median TRV to the minimum TRV was calculated and used as a measure of "spikiness" (or noisiness) of the data. The larger the TRV, the noisier the gene is. After examining the histogram of the TRV values across all genes (not shown), the threshold for tolerance level was set to 3 from the histogram of TRV values across all genes, and genes below the threshold were retained. These strict filtering rules enable the construction of a high-confidence dataset albeit possibly removing true signal. Finally, locally weighted scatter plot smoothing (LOWESS) was applied to further smooth the filtered data for robust estimation of kinetic parameters in the PECA model. To do so, the *lowess* function in R was used, a standard implementation of locally weighted scatter plot smoothing, with the default parameter settings. The extend of smoothing was manually inspected: The large majority of time-course profiles changed very little, except for those with zig-zag patterns between time points. Dataset EV4 shows the original and post-processed data. The final mRNA expression data for the two replicates are shown in Appendix Fig S5. As

described in the Results and Appendix, several tests, for example, comparison to RNA-sequencing data, validated the accuracy of the transcriptome data (Appendix Figs S6 and S7).

## Proteomics experiments

Cell pellets were collected for each sample and the cells were Dounce-homogenized in lysis buffer containing 10 mM KCl, 1.5 mM MgCl₂, 0.5 mM DTT, and 1X protease inhibitor cocktail (Complete, Mini, EDTA-free protease inhibitor cocktail tablets in EASYpack, Roche) in 10 mM Tris–HCl (pH 8.0). The samples were kept on ice throughout the entire procedure. Cell lysate was centrifuged at 1,000 × g at 4°C; the supernatant was saved as the cytosolic fraction, and the pellet was subjected to a single purification step via a sucrose cushion of 0.25 M and 0.88 M sucrose. The protein concentrations were determined using the Bradford protein assay (Bio-Rad) and the samples were diluted to 2 mg/ml concentration; 50 μl of each sample was mixed with equal volume of trifluoro-ethanol, then 15 mM DTT was added and incubated at 55°C for 45 min. Next, the samples were alkylated with 55 mM iodo-acetamide (IAA) for 30 min at room temperature in the dark. Then, the protein mixture was digested over night with mass spectro-metry-grade trypsin (Promega; at 1:50 v/w) at 37 °C. Tryptic digestion was halted by adding 2% formic acid (FA) and purified with C18 spintips (Thermo Scientific, HyperSep). The sample was stored at −80°C until LC-MS/MS analysis.

## LC-MS/MS analysis

Peptides were separated by reverse-phase nanoflow high-performance liquid chromatography (nano-HPLC) and quantitated on an LTQ Orbitrap Velos mass spectrometer (Thermo Scientific). Data-dependent analysis was performed at a resolution of 60,000 and with the top 20 most intense ions selected from each MS full scan, with dynamic exclusion set to 90 s if m/z acquisition was repeated within a 45-s interval. In each scan cycle, the top 20 fragmentation spectra were acquired in the collision-induced dissociation mode. For peptide separation, an Agilent ZORBAX 300SB-C18 reverse-phase column (150 mm × 75 μm inner diameter) and a 240-min gradient of 2–90% acetonitrile and 0.1% formic acid were used. The experiment was conducted twice (biological replicates) and four technical replicates for each sample were collected (repeat mass spectrometry measurements). The data for technical replicates were combined during the computational analysis, while the biological replicates were kept separately.

## Proteomics data processing and quality control

Raw data were processed using the MaxQuant software (1.3.0.3) (Cox and Mann, 2008), and peak lists were searched with Andro-meda (Cox *et al*, 2011) against a database containing the translation of all predicted proteins listed in Magrane (2011) and with a list of commonly observed contaminants supplied by MaxQuant. Protein identification was performed using 20 ppm tolerance at the MS level (FT mass analyzer) and 0.5 Da at the MS/MS level (ion trap analyzer), with a posterior global FDR of 1% based on the reverse sequence of the human FASTA file. Up to two missed trypsin cleav-ages were allowed, and oxidation of methionine and N-terminal

acetylation were searched as variable post-translational modification; cysteine carbamidomethylation as fixed modification. The minimal required peptide length was set to seven amino acids. The minimum number of peptide pairs used for quantitation was set to one. MaxQuant was used to combine the cytosolic and pellet samples as different fractions in the same experiment, after testing different options (Appendix Fig S8). Label-free quantitation (LFQ) with minimum ratio count set to 1 was used. Using only one peptide for quantitation can potentially lower quantitation accuracy; however, due to the time-series nature of the experiment, we were able to account for this additional variation by the examination of all data points across a time series and removal of noise. As is expected from complex proteomes such as that from mammalian cells, peptides can be shared between homologous proteins or splice variants, leading to "protein groups". The protein group structure is shown in MaxQuant's *proteinGroups.txt* file in the Dataset EV3; for clarity, we refer to the first and main protein from each group throughout the text.

A total of > 3,200 proteins were quantitated, of which 2,828 mapped to the RNA data. To derive a high-confidence dataset, all genes with one or more missing data points were removed, resulting in 1,820 genes for further processing. This dataset was normalized by the sum of all LFQ intensities, where the sum excluded the top 5% intensities in each sample. Removing the top 5% most intense proteins from the summation can prevent extremely abundant proteins (and potential outliers) from dominating the normalizing factor. The data were then log-transformed (natural log) and the jackknife procedure was applied to remove outliers (i.e., spikes) in the data, similar to what was described above for the transcriptomics data. The TRV threshold was 2 upon the examination of the histograms of all TRVs (not shown). Subsequently, locally weighted scatter plot smoothing (Lowess) was applied to further smooth the time-course data. A total of 1,237 genes were left in final dataset with complete and post-processed, high-confidence mRNA and protein annotations for two replicates across eight time points.

To evaluate the generality of our results, an extended dataset comprising 2,131 proteins was constructed. This dataset contains proteins with up to two missing values across the proteomics data which were imputed using Gaussian Processes. Appendix Fig S19 describes the details of this analysis and the results, which were consistent with those from the high-confidence dataset. The original "txt" folder of MaxQuant output files is provided as Dataset EV3. Several tests, for example, Western blotting, validated the quality of the proteomics data as described in the Results and Appendix (Appendix Figs S11 and S12).

### Hierarchical clustering and cluster analysis

Expression data were clustered using Perseus version 1.4.1.3. (http://141.61.102.17/perseus_doku), with default settings, that is, "correlation" and "average linkage" were used as the distance measures and clustering algorithm, respectively. Using a 0.604 distance threshold, the combined RNA and protein data were divided into 25 clusters. Clusters with more than 30 genes were chosen for function enrichment analysis using the NCBI DAVID tool (Huang *et al*, 2009a,b). Significantly enriched GO terms were chosen based on FDR < 0.05. The Dataset EV2 contains all GO term enrichments.

For further cluster analysis, 164 sequence features from various databases and tools were assembled (Appendix Table S2). These features include the characteristics that affect RNA and protein evolution, localization, synthesis, and degradation. Student's *t*-test and hypergeometric tests were conducted to calculate the enrichment of each sequence feature in any subset of genes. Bonferroni correction for multi-hypothesis testing was included to set the cutoff *P*-value.

### Protein expression control analysis

To quantitate the RNA- and protein-level expression regulation, protein expression control analysis (PECA) (Teo *et al*, 2014) was performed. As described in the original publication, PECA constructs a probabilistic model for the kinetic parameters governing the synthesis and degradation of an outcome molecule ($Y$) given the precursor molecule data ($X$). Specifically, the model estimates the ratio of synthesis and degradation over each time period given the data of $X$ and $Y$ at the beginning and end of each time period and also computes the posterior probability that the rate ratio has changed between two adjacent time periods given the paired data ($X$, $Y$) at three consecutive time points (two time periods). This probability, called the change point score (CPS), is then used to estimate the overall false discovery rate of regulation change events across all genes.

Since the raw data suggested that there are discordant expression patterns between the two biological replicates, we fitted the PECA model for each replicate separately. We then performed the RNA- and protein-level analyses separately. In the RNA-level analysis, we assumed that large-scale genomic changes such as those in ploidy have not occurred as a result of the ER stress within 30 h, and created an artificial DNA copy number data as the precursor molecule (variable $X$ in the PECA model), with the RNA data as the outcome (variable $Y$ in the model). We then estimated the ratio of the rates of RNA synthesis (transcription) and RNA degradation for each gene and computed the posterior probability that each intermediate time point is a change point where the rate ratio significantly changes (0.5, 1, 2, 8, 16, 24 h). In the protein-level analysis, we used RNA data as the precursor molecule ($X$) and protein data as the outcome ($Y$), where the ratio of translation and protein degradation was the kinetic parameter of interest for each protein along with their change point probability as described above.

To summarize the RNA-level and protein-level regulation changes across the three phases, we extracted the maximum CPS score in each phase and considered a gene as significantly regulated during the respective phase if the score was above the thresholds associated with FDR < 0.05. At this FDR, the CPS score thresholds for "significant" RNA- and protein-level regulation were 0.898 and 0.901 in the first replicate, respectively, and 0.890 and 0.898 in the second replicate. Genes were clustered using agglomerative clustering of the combined RNA/protein concentration data, followed by the dynamic tree cut algorithm (Langfelder *et al*, 2008). The six largest clusters as determined above were mapped to the data (Fig 5).

### Data availability

All transcriptomics data are deposited in the NCBI GEO database, with the identifier GSE67901. All proteomics data are publically

available from the ENSEMBL PRIDE database, identifier PXD002039.

**Expanded View** for this article is available online.

## Acknowledgements

We thank Rebecca Bish for help with the work. We acknowledge funding by the NYU Whitehead Fellowship (C.V.) and the NIH R01 GM113237 (C.V., Z.C., H.C., and G.T.). H.C. acknowledges funding by Singapore Ministry of Education Tier 2 grant (R-608-000-088-112).

## Author contributions

CV and HC perceived the idea of the study and designed the experiments. ZC and GT performed the experiments and analysis steps. SK, TMR, and HWLK contributed to parts of the study. ZC, GT, CV, and HC wrote the paper.

## Conflict of interest

The authors declare that they have no conflict of interest.

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
