## [Review Process File · Molecular Systems Biology]

Differential dynamics of the mammalian mRNA and protein expression in response to misfolding stress

Zhe Cheng, Guoshou Teo, Sabrina Krueger, Tara M. Rock, Hiromi W. L. Koh, Hyungwon Choi and Christine Vogel

Corresponding authors: Hyungwon Choi, National University Health System, Singapore and Christine Vogel, New York University

Review timeline:

Submission date:	04 July 2015
Editorial Decision:	12 August 2015
Revision received:	11 November 2015
Editorial Decision:	30 November 2015
Revision received:	04 December 2015
Accepted:	08 December 2015

Editor: Maria Polychronidou

Transaction Report:

1st Editorial Decision

12 August 2015

Thank you again for submitting your work to Molecular Systems Biology. We have now heard back from two of the three referees who agreed to evaluate your manuscript. Unfortunately, after several reminders we have not yet received a report from reviewer #1. Since the recommendations of the other two referees are quite similar, I prefer to make a decision now rather than further delaying the process. As you will see below, the referees raise a series of concerns, which should be carefully addressed in a revision of the manuscript. Their recommendations are rather clear so there is no need to repeat the points listed below.

If you feel you can satisfactorily deal with these points, you may wish to submit a revised version of your manuscript. Please attach a covering letter giving details of the way in which you have handled each of the points raised by the referees. A revised manuscript will be once again subject to review and you probably understand that we can give you no guarantee at this stage that the eventual outcome will be favorable.

REFeree COMMENTS

Reviewer #1

In the manuscript 'Differential dynamics of mammalian mRNA and protein expression in

response to misfolding stress' Teo et al. study joint timecourse proteomics and transcriptomics data in HeLa cells as response to stress induced by dithiothreitol. They apply a software tool called Protein Expression Control Analysis (PECA) to this data, which was produced and already published by the same authors.

The findings are well presented in the manuscript and the authors make a strong case for that at least in the system that they studied the posttranscriptional regulation is as important as the transcriptional regulation

The computational methods appear to be simple and robust which is an important aspect for comparability of the results between different studies and systems.

The authors come to novel insights regarding the relationship between transcript and protein regulation that are worth reporting in MSB.

In summary I recommend the acceptance for publication.

Reviewer #2:

Summary

The manuscript of Cheng et al from Vogel group is an interesting study in which the authors address an important question in gene expression, i.e., the relative importance of regulation at mRNA versus protein level in an induced response. Specifically, they acquired longitudinal label-free transcriptomic and proteomic data from HeLa cells that were treated by DTT to induce ER stress. They applied their previously described PECA software to extract the rate profiles of both mRNAs and proteins during the time course. According to the PECA results (profiles, the number of genes significantly regulated at each time points, etc.) they found "spike-like" patterns in mRNA dynamics and "switch-like" patterns in protein dynamics in their system. They further conclude that both regulatory levels are "equally" important but mRNA fold changes were much smaller than those of proteins.

General remarks

Generally this is a well-conducted study. Following the recent important literature such as Jovanovic et al 2015 and Battle et al 2015 at Science and Selbach paper at Nature in 2011, this paper continues the debate of mRNA/protein importance, and will hopefully contribute to the deeper understanding of gene expression regulation in mammalian cells by providing another valuable data resource to the community.

This being said, the current version of the study will benefit significantly from the careful generalization of their conclusion and the further analysis of some technical problems (see below). Further, the biological principles, especially those novel ones discovered by this study, should be summarized in a more informative fashion.

Major points

1. The first concern is how to generalize their conclusion from this specific study and position it into the previous knowledge established by other studies. ER stress normally leads to protein misfolding and degradation, a process in which post-translational regulation is already known to be very important. This system could be distinctive from other typical signaling systems (such as LPS treatment in Jovanovic et al). In fact, in Figure S17 where data of Jovanovic et al is analyzed by PECA, the RNA fold change tends to be as high as protein fold change (based on heatmaps in Figure S17A) and generally follows a quite different pattern compared to Figure 2 in this study. The authors, instead of only discussing the similar aspects using descriptive words (Page 7, Para 4), should carefully analyze the difference between systems in the quantitative way and thoroughly discuss the boundaries of this present study. The similarities and differences should be discussed in the biological context of the respective stimuli used.

2. A second significant concern is the dynamic range of detected proteins. The authors detected 3,235 proteins in total but only managed to quantify 1,237 proteins across samples, which is only

54% of the number analyzed by in Jovanovic et al. Even though, the authors claim that they have a comparable number to the previous study. The number of proteins and the significance of these numbers should be discussed in view of the published literature. More importantly, as noted by previous papers, the absolute protein quantity (copies per cell) is an important feature for mediating protein expression. How to extend the observations of the top 1000 abundant proteins in this study to the rest proteome (other estimated >9000 proteins expressed in a cell)? The authors indicated that the mRNA pattern might remain the same in this context (Figure S16), but they would need to perform further experiments or analyses to show this is also the case for the proteome.

3. The author noted that a portion of cells underwent apoptosis in the early phase. Gene expression may be halted or permanently stopped in the apoptotic or dead cells. How to estimate the potential bias resulted? Minimally the fraction of apoptotic cells should be measured and indicated at each time point.

4. Based on the comparison of label-free proteomic data and RNA-seq data, the authors conclude that the magnitude of change is much more pronounced for proteins than mRNAs. However this is a problematic conclusion because the quantitative variation in different proteomic approaches is quite different. E.g., the SILAC experiment, or the MS2 level quantification by DIA or SRM could be quantitatively more precise than LFQ MS1 profiling or spectral counting. This could greatly affect the overall fold change observed. Just for reference, in Figure S17 where SILAC data is used, the fold-change of proteins seems to be quite similar to that of transcripts. The authors are requested to further explore the quantitative accuracy and dynamic range of the quantification method used.

5. Their biological investigation overall is descriptive-accordingly the subtitles of Results are presented just as "Per-gene regulation" or "Example patterns". Also, lots of biological insights were put into Supplementary. The manuscript would be greatly improved if biological observations, especially those novel ones, were summarized in an organic and logical manner (together with e.g., Figure S14, S17 etc.) with informative subtitles.

6. The description of the "buffering" concept needs to be revised. The "opposite direction" of regulations is NOT necessarily a "buffering" (Figure 3; Page 6, Para 3; Figure S11). "Buffering", in essence, describes the reduction of variations or differences from the 1st level, thus conferring robustness or stability at the 2nd level (e.g., please see Bader et al 2015, *Molecular Systems Biology*). The other description of "buffering" by the authors (Page 8 Para1) is in the same line. Please refine the corresponding text and results.

Minor points

1. The design of "backwards" culturing method is not explained in the Results, making the understanding of Figure 1A (upper panel, especially the time points of x-axis) extremely difficult. Please add another illustration here.

2. The classification of early, intermediate and late phases is a bit arbitrary. A PCA analysis, for example, will be beneficial to support the classification. Also all the p-values of the enriched processes in each phase should be given to Figure 1B and Page 5 Para 3.

3. The correlation plot between mRNA and protein quantities at different time points would be very informative for general readers.

4. The limitations of PECA and the present study approach need to be fairly discussed and compared to other models and other approaches, e.g., pSILAC (which clearly has the advantage of separating newly synthesized proteins and protein degradation).

5. The order of Supplementary Figures cited in the text is not correct in many cases.

6. The order of Figure S8 B and C legend is wrong and should be switched.

7. The western blot results in Figure S8 do not fully support the LFQ results (e.g., A burst of ATP5D protein expression at 2nd hour is not confirmed by the western blot). The relevant PECA results should also be displayed.

8. The number of quantified proteins is not consistent (1,266 in Methods, Page 13 Para 3, versus 1,237 in Page 5 Para 3). Also the final sentence in Page 5 Para 3 is unclear.

Reviewer #3:

This manuscript compares the temporal changes in the transcriptome and proteome of HeLa cells following incubation with dithiothreitol (DTT). The authors performed transcriptomics and label-free quantitative proteomics analyses of cell extracts from 8 time points ranging from 0-30 h post DTT treatment. To profile changes in gene/protein regulation and identify genes and time points at which these changes are taking place, data were processed with Protein Expression Control Analysis (PECA), statistical analysis software recently developed by their group (Teo et al., *J Proteome Res.*, 13, 29, 2014). A salient finding of the present study is the observation that DTT resulted in early transcription of several genes including those associated with apoptosis and ER-stress response, whereas changes in protein levels were more continuous and reflected gradual changes in protein synthesis and degradation. Such data is potentially rich in novel information and could provide valuable insights on the dynamics of cell response to protein misfolding. These datasets are the core of the paper and its main strength. The use of PECA to compare mRNA and protein expression levels provides a framework to generate interesting predictions (e.g. ER stress, redox imbalance and mitochondria response; regulation of aminoacyl-tRNA synthetases), but none of these are not further verified. There are also some concerns regarding the data analysis and the way the work has been presented in the manuscript. Specific comments are provided below.

1. On p. 5 authors write "In sum, while suffering from a loss of cells during the early phase of the experiment, the surviving cell population continued division throughout the entire time course." and illustrate this in figure 1. The top panel of the figure (cell counting) is not entirely intuitive. How come cell number drops even w/o DTT treatment (left side, T0). Why after 2 h treatment cell number still grows and only later starts to decrease? Why on the right side (30h) cell number decreases and then increases (even before the treatment) and then decrease again (right after the treatment). What does a DTT half-life of 4 h means in the context of prolonged exposure (30 h)?

2. On p. 5, the section on "Dynamics of the integrated transcriptome and proteome" requires further details on data analysis. The authors should provide statistical analyses on total number of quantified proteins (with corresponding #peptides) across the number of time points. Did they use only peptides unique to proteins for quantification? In the methods section they indicated: "We used label free quantification (LFQ) with minimum ratio count set to 1". This suggests they performed protein quantification based on a single peptide which is not a generally accepted as a common practice. In supplementary data they indicated that: "The experiment was conducted in a triplicate, and two replicates of higher data quality were selected for further analysis." Are proteomics data generated from two or three biological (or technical) replicates? How was sample normalization performed (p.13)? Did the authors use protein quantification or same number of cells for time point? It would be pertinent to display the abundance profile (from MS data) of specific housekeeping proteins (e.g. actin, tubulin) to illustrate that they remain constant throughout the experiment.

3. In the method section, the authors should add information on MS measurements and data analysis. Particularly, the fragmentation mode (CID/HCD, ...), normalized collision energy, dynamic exclusion, etc. It would be also beneficial to add MaxQuant output tables as a supplementary data so the readers could easily access all search parameters (parameters.txt) and results (summary.txt, evidence.txt and proteinGroups.txt). Many questions related to data analysis could be answered by providing Perseus session file (with .sps extension) as it contains all data analysis steps.

4. On p. 5 (and method section p. 14), the authors should provide additional information on how they compare kinetic profiles of individual gene/protein from transcriptomics/proteomics datasets and how clusters were selected to identify specific trends (e.g. concordant or discordant clusters between omics datasets).

5. Figures S4 and S7 show good correlation between individual measurements. However, the comparison of dynamic trends (e.g. effect of "buffering" on figure S11) is less consistent. Why the overlap between proteins with "buffering effect" (see Dataset1 excel file, tab "Buffering") is so

small (e.g. 52 genes in R1, 59 genes in R2 and only 10 genes common to both R1 and R2)? Since PECA analyzes temporal profiles of mRNA/protein it would be important to see a good correlation of temporal patterns between replicates, a point that has not been addressed in the manuscript. The lack of consistency between replicates is also evident from Figures 2 and S9. The authors should display protein/mRNA expression temporal profiles for all "buffering" genes to compare profiles between replicates. The only cases presented are for splice variants in QARS (Fig S14), where trends are similar though the shapes of the temporal profiles are different. Since PECA analysis is using the shape of the temporal profile rather than just a up/down trend, these do not appear to be representative examples of good correlation between replicates.

6. On p. 7, p. 9 and several other places throughout the text. The term "switch-like manner" could be misleading considering time resolution of the experiment, if the authors observe change only between 2 time points this does not necessarily imply a switch-like manner. Using higher temporal resolution might reveal slow steady-state changes instead.

7. Dataset1 table contains individual proteins in separate rows. In the proteomics experiments several peptides can be shared among the homologous proteins, and MaxQuant reports them as a protein group. How was this information reported in the table?

8. On p. 12 "We devised a jackknife procedure to remove aberrant expression measurements". I'm not sure if this procedure is applicable for the selected time frame (e.g. time points: 0, 0.5, 1, 2, 8, 16, 24 and 30 h). I would understand if the measurements were performed every 5 min, and a sudden spike is probably a technical artifact. Under the present condition, an increase in abundance between 8 to 16 h followed by a decrease to original level at 24h would be interpreted as a spike, a situation that might not be realistic for these time points.

9. On p. 13 "Finally, locally weighted scatter plot smoothing (lowess) was applied to further smooth the filtered data for robust estimation of kinetic parameters in the PECA model." More details are required. With lowess one can smooth any profile even if the actual data look terrible, what were the parameters used for the smoothing and which level of RSD was considered as acceptable?

10. Proteomics experiments (p.13). Was there a rationale for separating nuclear and cytosolic fractions since no comparison was performed between fractions?

Minor comments.

1. Figure S10, figure legends (Protein-1/Protein-2) are not explained in the text.
2. Description in the file Dataset 1 needs more details (e.g., are numbers in the column "Protein replicate 1" represent intensity, what do the column "Protein rel rep1" correspond to?)
3. Check text for consistency (e.g. p. 13 "Each sample was by with three to four" something is missing here).

1st Revision - authors' response

11 November 2015

Detailed responses

Our replies are printed in bold font.

Reviewer #1:

In the manuscript 'Differential dynamics of mammalian mRNA and protein expression in response to misfolding stress' Teo et al. study joint timecourse proteomics and transcriptomics data in HeLa cells as response to stress induced by dithiothreitol. They apply a software tool called Protein Expression Control Analysis (PECA) to this data, which was produced and already published by the same authors.

The findings are well presented in the manuscript and the authors make a strong case for that at least in the system that they studied the posttranscriptional regulation is as important as the transcriptional regulation

The computational methods appear to be simple and robust which is an important aspect for comparability of the results between different studies and systems.

The authors come to novel insights regarding the relationship between transcript and protein regulation that are worth reporting in MSB.

In summary I recommend the acceptance for publication.

Reviewer #2:

Summary

The manuscript of Cheng et al from Vogel group is an interesting study in which the authors address an important question in gene expression, i.e., the relative importance of regulation at mRNA versus protein level in an induced response. Specifically, they acquired longitudinal label-free transcriptomic and proteomic data from HeLa cells that were treated by DTT to induce ER stress. They applied their previously described PECA software to extract the rate profiles of both mRNAs and proteins during the time course. According to the PECA results (profiles, the number of genes significantly regulated at each time points, etc.) they found "spike-like" patterns in mRNA dynamics and "switch-like" patterns in protein dynamics in their system. They further conclude that both regulatory levels are "equally" important but mRNA fold changes were much smaller than those of proteins.

General remarks

Generally this is a well-conducted study. Following the recent important literature such as Jovanovic et al 2015 and Battle et al 2015 at Science and Selbach paper at Nature in 2011, this paper continues the debate of mRNA/protein importance, and will hopefully contribute to the deeper understanding of gene expression regulation in mammalian cells by providing another valuable data resource to the community.

This being said, the current version of the study will benefit significantly from the careful generalization of their conclusion and the further analysis of some technical problems (see below). Further, the biological principles, especially those novel ones discovered by this study, should be summarized in a more informative fashion.

Based on the reviewer's comments, we substantially extended the comparison of our findings with those by Jovanovic et al. (Science 2015). This comparison and a discussion of the generality of our findings has been worked into the revised Discussion section. We further edited figure legends, Results, Discussion section, and the Abstract to better present our novel findings and clarify any misunderstandings about technical aspects. The details of these revisions are specified below next to the reviewer's comments.

Major points

1. The first concern is how to generalize their conclusion from this specific study and position it into the previous knowledge established by other studies. ER stress normally leads to protein misfolding and degradation, a process in which post-translational regulation is already known to be very important. This system could be distinctive from other typical signaling systems (such as LPS treatment in Jovanovic et al). In fact, in Figure S17 where data of Jovanovic et al is analyzed by PECA, the RNA fold change tends to be as high as protein fold change (based on heatmaps in Figure S17A) and generally follows a quite different pattern compared to Figure 2 in this study. The authors, instead of only discussing the similar aspects using descriptive words (Page 7, Para 4), should carefully analyze the difference between systems in the quantitative way and thoroughly

discuss the boundaries of this present study. The similarities and differences should be discussed in the biological context of the respective stimuli used.

The revised manuscript now contains Table S1 in the Appendix (Notes) file which compares the average fold changes in this and the Jovanovic study. It also contains, for the Jovanovic study, an analysis similar to that presented in Figure 5 of the manuscript in which we analyze major temporal patterns (Appendix Figure S23). These additional analyses enabled a thorough and more quantitative comparison between the two systems.

The revised Discussion section now has an expanded comparison with the Jovanovic study, highlighting both similarities and differences (p. 12, 13):

“Overall, the transcriptome in our dataset was comparatively static, consistent with earlier observations (Murray et al, 2004). In contrast, as expected from a treatment that affects the protein homeostasis function of the ER, we found that protein concentrations changed more drastically than those of mRNA. However, despite the smaller RNA concentration changes, we found that mRNA- and protein-level regulation contributed equally to the final expression response (Table 1). This finding contrasts reports from steady-state systems in yeast (Csardi et al, 2015a; Li et al, 2014a), but is consistent with a recent study on a dynamic system of mammalian cells responding to lipopolysaccharide (LPS) treatment (Jovanovic et al, 2015). In both the ER stress response described here and the published data on the LPS response, regulation at both RNA and protein level contributes to the change in the system, and protein expression changes drive the synthesis and turnover of highly abundant molecules (Appendix Figures S22, S23, Tables S1, S3, S4). ”

...

“We aimed to estimate the generality of these findings by re-analyzing the published dataset on the LPS response with PECA (Appendix Figure S22, S23, Tables S3, S4)(Jovanovic et al, 2015). This re-analysis confirmed that the LPS response is driven by substantial RNA regulation immediately after stimulation (Jovanovic et al, 2015). However, we also found that the changes in protein concentrations were not entirely accounted for by RNA regulation alone: the rates of translation and protein degradation also changed significantly and fine-tuned the final protein concentrations. Again, we observed a spike- and switch-like behavior similar to that of the ER stress response, suggesting transient and permanent regulatory changes, respectively. However, in contrast to the ER stress response in which proteins appeared to switch from the original to a new steady state, in the LPS response we found the switch-like behavior for RNA-level regulation. Given that the LPS response is driven largely by transcription, while the ER stress response strongly affects the proteome, we hypothesize that the switch-like regulatory patterns occur in the dominant level of regulation to foster major, and perhaps permanent changes of the cellular state. Future work will have to test the validity of this hypothesis.”

2. A second significant concern is the dynamic range of detected proteins. The authors detected 3,235 proteins in total but only managed to quantify 1,237 proteins across samples, which is only 54% of the number analyzed by in Jovanovic et al. Even though, the authors claim that they have a comparable number to the previous study. The number of proteins and the significance of these numbers should be discussed in view of the published literature. More importantly, as noted by previous papers, the absolute protein quantity (copies per cell) is an important feature for mediating protein expression. How to extend the observations of the top 1000 abundant proteins in this study to the rest proteome (other estimated >9000 proteins expressed in a cell)? The authors indicated that the mRNA pattern might remain the same in this context (Figure S16), but they would need to perform further experiments or analyses to show this is also the case for the proteome.

The dynamic range of the protein concentrations covers about five orders of magnitude (Appendix Table S1) which is similar to other large-scale proteomics studies (e.g. Selbach, Nature 2011). While analyses exist that detect >11,000 proteins in a single sample, detection of a specific protein across all time points in a time series

experiment and in all replicates reduces coverage dramatically, which is the reason for the fact that even the most recent study by Jovanovic et al. covers ‘only’ 2,288 proteins in the complete dataset.

The revised manuscript now presents the results for an extended dataset in which we analyze 2,131 proteins (which represents 93% of the size of the Jovanovic data). This dataset is noisier, but shows similar results with respect to the temporal patterns of regulation (Appendix Figure S19).

To ensure reliability of our findings, we decided to focus our analyses on the high-confidence dataset, as it has, albeit being smaller, replicate measurements for all proteins and no missing data points. We now explicitly state so, but also mention the limitations of this approach. We discuss these considerations in the revised manuscript:

Results, p. 7 “Protein concentrations span about five orders of magnitude in concentrations (Appendix Table S1) which is similar to what other large-scale studies observe (Schwanhausser et al, 2011).”

Discussion, p. 12 “While the major results from this study have been confirmed by analysis of the total transcriptome (Appendix Figure S21) and an extended proteomics dataset (Appendix Figure S19), our discussions are still restricted to a comparatively small subset of the proteome. However, the use of complete time series data and biological replicates support our confidence in the validity of our findings.”

3. The author noted that a portion of cells underwent apoptosis in the early phase. Gene expression may be halted or permanently stopped in the apoptotic or dead cells. How to estimate the potential bias resulted? Minimally the fraction of apoptotic cells should be measured and indicated at each time point.

We amended Figure S3 in the Appendix to quantify the amount of apoptotic cells as measured in the FACS experiment. These numbers indicate that apoptosis affected up to 45% of the cells, but only at one time point (2 hours). Other time points have much lower estimates. Since dead cells and debris is immediately lost during the sample prep, the RNA and protein concentration data focusses largely on life cells. We now discuss these considerations in the Appendix and the manuscript:

Results, p. 6 “This interpretation was confirmed by assays monitoring cell cycle progression and apoptosis: while apoptosis occurred during the first two hours of the experiment, later time points showed continued division of the majority cells (Figure 1A, middle/lower panel; Appendix Figure S3). DNA-labeling coupled to flow cytometry showed that apoptosis peaked at 2 hours, with ~45% of cell death. Notably, the sample preparation for the mRNA and protein analysis discarded cellular debris; the results below hence focus on life cells.”

4. Based on the comparison of label-free proteomic data and RNA-seq data, the authors conclude that the magnitude of change is much more pronounced for proteins than mRNAs. However this is a problematic conclusion because the quantitative variation in different proteomic approaches is quite different. E.g., the SILAC experiment, or the MS2 level quantification by DIA or SRM could be quantitatively more precise than LFQ MS1 profiling or spectral counting. This could greatly affect the overall fold change observed. Just for reference, in Figure S17 where SILAC data is used, the fold-change of proteins seems to be quite similar to that of transcripts. The authors are requested to further explore the quantitative accuracy and dynamic range of the quantification method used.

To address the reviewer’s comments, we extended the Appendix Notes by several items: Table S1 which lists the fold-changes and dynamic ranges in expression for our and Jovanovic et al.’s data; and Figures S6 and S10 which show the correlations between replicate measurements for the RNA and protein data respectively, including

correlation coefficients and a table with average fold-changes between replicates (and standard deviations).

The reviewer is correct in that the ‘biological’ fold-changes during the LPS response are similar between RNA and protein data (Table S1). However, as was discussed above (also by this reviewer), this finding might largely be due to the fact that the LPS response is driven by the transcriptome response. We compare the Jovanovic and our data extensively in the revised Discussion (p. 12, 13, and see above).

Further, as noted, the ER stress response shows much larger concentration fold-changes at the protein level compared to the RNA level, despite similar dynamic range of detection (Table S1) which is five orders of magnitude. This result is expected from a system that affects proteostasis, such as that of the ER stress response.

In addition, these ‘biological’ fold changes are larger than the ‘technical’ fold-changes detected between replicate measurements which are detailed in Figures S6 and S10: the Pearson correlation coefficient between replicates is $R > 0.97$ and $R > 0.94$ for the RNA and protein concentrations, respectively (except for time=30 hours with $R = 0.88$). The average fold-changes between replicate measurements are centered around 1 (no change) for both mRNA and protein, but the spread is wider for protein (standard deviation ~ 1 and ~ 2 for mRNA and protein, respectively). (Averages were calculated on the log-transformed ratios.)

The high reproducibility between replicate measurements increases our confidence in accurate identification of expression fold-changes. However, we conducted additional tests (i.e. comparison with RNA-seq data, western blots of individual proteins, and examination of housekeeping genes, see below) to further validate the quantitative measurements.

We discuss the reproducibility and accuracy of the quantitative data and the different forms of validation in the revised manuscript (p. 6/7, 16) and in the Appendix Notes (Figures S6, S7, S10, S11, S12). Finally, as mentioned above, we focus on the dataset with complete time series data across both replicates to ensure high-accuracy measurements and reliability of our findings. (Discussion, p. 12 “However, the use of complete time series data and biological replicates support our confidence in the validity of our findings.”)

Please also see the discussion of the ‘RCM’ measure in response to Reviewer #3’s comments (below).

5. Their biological investigation overall is descriptive-accordingly the subtitles of Results are presented just as "Per-gene regulation" or "Example patterns". Also, lots of biological insights were put into Supplementary. The manuscript would be greatly improved if biological observations, especially those novel ones, were summarized in an organic and logical manner (together with e.g., Figure S14, S17 etc.) with informative subtitles.

We revised the main text and Appendix Notes thoroughly with respect to the reviewer’s comments and changed figure/table titles as well as headings to more informative titles. Further, we edited the entire Results and Discussion part of the main text to improve the logical flow and link different parts with each other. Main findings are summarized in the Abstract, the Standfirst text, at the beginning and end of the sections.

In addition, we moved the example of the chaperones to the main text (Results, p. 10, 11) and extended the discussion of the aminoacyl-tRNA synthetases (p. 11, 12).

Finally, we hope that the extended comparison with the Jovanovic dataset (see above) will help placing our new findings into a wider biological context.

6. The description of the "buffering" concept needs to be revised. The "opposite direction" of regulations is NOT necessarily a "buffering" (Figure 3; Page 6, Para 3; Figure S11). "Buffering", in essence, describes the reduction of variations or differences from the 1st level, thus conferring robustness or stability at the 2nd level (e.g., please see Bader et al 2015, Molecular Systems Biology). The other description of "buffering" by the authors (Page 8 Para1) is in the same line. Please refine the corresponding text and results.

We revised this part of the Results and Discussion substantially and now distinguish between regulation in opposite direction (discordance) and actual buffering. We place the emphasis on cases of discordant regulation which are rare in our dataset. The revised manuscript discusses these points in Appendix Figure S16 and the main text.

Results, p. 9 “Next, we asked if mRNA- and protein-level regulation occurred in a concordant fashion, i.e. in the same direction, or discordantly, i.e. working in opposing directions. One such example is GRP78 (Figure 3) for which mRNA expression is down-regulated and protein expression is up-regulated at the 16-hour time point. Table 1 already indicates that discordant regulation is comparatively rare: only few genes are listed in the lower left and upper right corners of the tables (75 genes in total). Alternative ways to identify discordant regulation confirmed this result, i.e. via filtering for negative correlation between PECA’s mRNA and protein rate ratios across both replicates (Appendix Dataset 1, Appendix Figure S16A,B). We then further refined this analysis and required not only opposing regulation, i.e. at least one significant regulatory event at the mRNA and one at the protein level, but also constant protein concentrations, i.e. changes smaller than 1.5-fold across both biological replicates. Such a scenario would indicate cases of ‘buffering’ in which changes in mRNA concentrations are counterbalanced to result in no overall change at the protein level. Three out of the 75 genes passed this additional filtering and are shown in Appendix Figure S16C. One of these genes is HSC70 (RCM=0.91/0.09), a chaperone discussed below (Figure 6A). ... Overall, we conclude that discordant regulation is rare, and most changes occur in coordinated manner between mRNA and protein expression.”

Discussion, p. 13 “While most changes in concentrations and rate ratios were concordant between mRNA and protein, we also asked if we observed discordance in the dataset, i.e. case in which transcript and protein level regulation act in opposite directions. Some examples of such discordance can be identified in Table 1, and we found that they were rare. One example is the ER stress related chaperone GRP78, whose mRNA concentration is in decline at 16 hours, while protein concentrations still increase (Figure 3). We find that discordance at a specific time point is often resolved by a simple delay in the response: the changes at the RNA level are initially counteracted by the protein level response, but later supported by concordant action.”

Minor points

1. The design of "backwards" culturing method is not explained in the Results, making the understanding of Figure 1A (upper panel, especially the time points of x -axis) extremely difficult. Please add another illustration here.

We revised Figure 1A to include more annotations and a clearer setup, and we edited the Results section accordingly (p. 6). We also added a figure to the Appendix that explains the experimental design (Figure S1).

To avoid confusion, we removed the word “backwards” and rewrote the respective description to “At approximately 60% confluency, dithiothreitol (DTT) at 2.5 mM concentration was added to induce stress for different periods of time, i.e. 0, 0.5, 1, 2, 8, 16, 24, and 30 hours. To account for different cellular ages at harvest, the experiment was conducted so that the experiment started with the 30h treatment

period, and the cells were then collected at the same time (Appendix Figure S1).” (p. 13).

2. The classification of early, intermediate and late phases is a bit arbitrary. A PCA analysis, for example, will be beneficial to support the classification. Also all the p-values of the enriched processes in each phase should be given to Figure 1B and Page 5 Para 3.

We added significance measures (FDR and p-values) to Figure 1B and the text were appropriate.

We also performed PCA analysis which is shown in Figure S4. Its results show that the intermediate phase represents a transition between the clearly distinct early and late phase of the experiment. However, we felt that the distinction of the three phases plays only a minor role in the paper, and therefore we kept its discussion in the main text to a minimum. The new Figure S4 is now mentioned in the Results, p. 7.

3. The correlation plot between mRNA and protein quantities at different time points would be very informative for general readers.

The Appendix Material now contains a figure with all correlations between protein and mRNA at all time points (Figure S13), which is also mentioned in the main text, p. 8.

4. The limitations of PECA and the present study approach need to be fairly discussed and compared to other models and other approaches, e.g., pSILAC (which clearly has the advantage of separating newly synthesized proteins and protein degradation).

We extended the discussion of PECA’s advantages and disadvantages on p. 4/5: “Compared to experimental measurements of protein synthesis and degradation rates, e.g. pulsed and dynamic SILAC (Doherty et al, 2009; Schwanhausser et al, 2009), PECA has the disadvantage that it currently does not distinguish between molecular synthesis and degradation, but the advantage that it does not require metabolic labeling of the proteins, and can therefore be applied to systems that are not amenable to SILAC. Label-free proteomics approaches are slightly less accurate than those using isotopic labeling, and therefore cannot detect small fold-changes as sensitively. However, this disadvantage is effectively compensated for by recent technological and computational advances, and easier sample handling that allows for analysis of multiple replicates (Cox et al, 2014; Liu et al, 2013; Schmidt et al, 2014; Tebbe et al, 2015).

Although a few other computational approaches can quantify the rate parameters based on first-order differential equations, e.g. (Jovanovic et al, 2015; Lee et al, 2011; Omranian et al, 2015), PECA is the first approach that introduced a probabilistic model for statistical inference of regulatory parameters. Unlike the alternative approaches, PECA’s probabilistic model is formulated based on Bayesian hierarchical models and leads to comparatively stable parameter estimation. More importantly, it provides a statistical score, called change point probability score (CPS), on which one can apply a score threshold associated with a desired false discovery rate (FDR) to extract genes that are significantly regulated at one or both levels. ‘Significant regulation’ can therefore be defined as a significant change in the rates of synthesis and degradation of a gene between consecutive time intervals. The ability to estimate FDRs provides a unified analysis framework to identify mRNA- and protein-level regulation above the noise level.”

5. The order of Supplementary Figures cited in the text is not correct in many cases.

The numbering has been corrected.

6. The order of Figure S8 B and C legend is wrong and should be switched.

Appendix Figure S11 (was Figure S8) has been edited accordingly.

7. The western blot results in Figure S8 do not fully support the LFQ results (e.g., A burst of ATP5D protein expression at 2nd hour is not confirmed by the western blot). The relevant PECA results should also be displayed.

Appendix Figure S11 (was Figure S8) has been amended accordingly to also show the PECA results for the genes. We note, however, that the western blotting aims to confirm protein concentration measurements; in contrast, PECA transforms these protein concentration measurements (by incorporating mRNA concentration changes) into information on regulatory information. Hence some differences between the protein concentrations and the PECA rate ratio profiles can be observed. We added a note to the figure legend.

8. The number of quantified proteins is not consistent (1,266 in Methods, Page 13 Para 3, versus 1,237 in Page 5 Para 3). Also the final sentence in Page 5 Para 3 is unclear.

The correct number is 1,237. We fixed this typo and clarified the last sentence on p. 5/para. 3 (now p. 6) to “The increase in lysosomal proteins is consistent with observations which found that the UPR remodels the lysosome as part of a pro-survival response (Brewer et al, 2008; Elfrink et al, 2013; Ron & Hampton, 2004; Sriburi et al, 2004).”

Reviewer #3:

This manuscript compares the temporal changes in the transcriptome and proteome of HeLa cells following incubation with dithiothreitol (DTT). The authors performed transcriptomics and label-free quantitative proteomics analyses of cell extracts from 8 time points ranging from 0-30 h post DTT treatment. To profile changes in gene/protein regulation and identify genes and time points at which these changes are taking place, data were processed with Protein Expression Control Analysis (PECA), statistical analysis software recently developed by their group (Teo et al., *J Proteome Res.*, 13, 29, 2014). A salient finding of the present study is the observation that DTT resulted in early transcription of several genes including those associated with apoptosis and ER-stress response, whereas changes in protein levels were more continuous and reflected gradual changes in protein synthesis and degradation. Such data is potentially rich in novel information and could provide valuable insights on the dynamics of cell response to protein misfolding. These datasets are the core of the paper and its main strength. The use of PECA to compare mRNA and protein expression levels provides a framework to generate interesting predictions (e.g. ER stress, redox imbalance and mitochondria response; regulation of aminoacyl-tRNA synthetases), but none of these are not further verified. There are also some concerns regarding the data analysis and the way the work has been presented in the manuscript. Specific comments are provided below.

We hope that the revisions described above and below have clarified the misunderstandings and convince this reviewer of the quality of the work.

1. On p. 5 authors write "In sum, while suffering from a loss of cells during the early phase of the experiment, the surviving cell population continued division throughout the entire time course." and illustrate this in figure 1. The top panel of the figure (cell counting) is not entirely intuitive. How come cell number drops even w/o DTT treatment (left side, T0). Why after 2 h treatment cell number still grows and only later starts to decrease? Why on the right side (30h) cell number decreases and then increases (even before the treatment) and then decrease again (right after the treatment). What does a DTT half-life of 4 h means in the context of prolonged exposure (30 h)?

The reviewer’s questions originate from a misunderstanding of Figure 1A. For example, cell numbers do not drop without DTT treatment, and cells continued to divide as is illustrated by an increase in cell counts and the presence of a mitotic marker.

In the light of this Reviewer's and Reviewer #2's comments, we revised Figure 1A substantially to illustrate the experimental approach more clearly. In addition, we added Appendix Figure S1 to explain the experimental setup further.

We also added a discussion of the DTT half-life to the main text, page 5: "In this setup, DTT had a half-life of ~4h (Appendix Figure S2)."

2. On p. 5, the section on "Dynamics of the integrated transcriptome and proteome" requires further details on data analysis. The authors should provide statistical analyses on total number of quantified proteins (with corresponding #peptides) across the number of time points. Did they use only peptides unique to proteins for quantification? In the methods section they indicated: "We used label free quantification (LFQ) with minimum ratio count set to 1". This suggests they performed protein quantification based on a single peptide which is not a generally accepted as a common practice. In supplementary data they indicated that: "The experiment was conducted in a triplicate, and two replicates of higher data quality were selected for further analysis." Are proteomics data generated from two or three biological (or technical) replicates? How was sample normalization performed (p.13)? Did the authors use protein quantification or same number of cells for time point? It would be pertinent to display the abundance profile (from MS data) of specific housekeeping proteins (e.g. actin, tubulin) to illustrate that they remain constant throughout the experiment.

In the light of these questions and those below, we extended the description of the mass spec data collection and analysis. Specifically:

We added Appendix Dataset 3 which contains all output files from the MaxQuant software which includes a file called *peptides.txt* with the peptides that were used for quantitation. We note that the PECA analysis has been conducted at the protein level (after evidence from multiple peptides had been combined), but our future (unpublished) developments of PECA also include a peptide-level analysis. The *proteinGroups.txt* and the *summary.txt* files contain the summary on total numbers of proteins and peptides identified in the experiment.

Based on the reviewer's comment we also expanded the discussion on why we set the LFQ minimum ratio count to 1. On p. 17 we write "We used label free quantification (LFQ) with minimum ratio count set to 1. Using only one peptide for quantitation can potentially lower quantitation accuracy; however, due to the time-series nature of the experiment, we were able to account for this additional variation by examination of all data points across a time series and removal of noise (see below)." Measurement noise (and inaccuracies) were removed by outlier detection (and removal) and also light smoothing, as described in the Methods sections, p. 16, 17, and below.

We clarified the distinction between biological replicates (two) and technical replicates (four) per sample. On p. 17 we write "We conducted the experiment twice (biological replicates) and collected four technical replicates for each sample (repeat mass spectrometry measurements). The data for technical replicates were combined during the computational analysis (see below), while the biological replicates were kept separately."

Further, we extended the description of the sample normalization (of the RNA and proteomics data). On p. 16 we write (for RNA data processing): "We devised a jackknife procedure to remove aberrant expression measurements. In the procedure, one data point was removed at a time and the total range of variation (TRV), defined as the difference between maximum and minimum, for the remaining dataset was recorded. After following this procedure for all data points, the ratio of the median TRV to the minimum TRV was calculated and used as a measure of 'spikiness' (or noisiness) of the data. The larger the TRV, the noisier the gene. After examining the histogram of the TRV values across all genes (not shown), the threshold for tolerance level was set to 3 from the histogram of TRV values across all genes, and genes below the threshold were retained. These strict filtering rules enable construction of a high-confidence dataset albeit possibly removing true signal. Finally, locally weighted scatter plot smoothing (Lowess) was applied to further smooth the

filtered data for robust estimation of kinetic parameters in the PECA model. To do so, we used the lowess function in R, a standard implementation of local weighted scatter plot smoothing, with the default parameter settings. We manually inspected the extend of smoothing: the large majority of time course profiles changed very little, except for those with zig-zag patterns between time points. Appendix Dataset 4 shows the original and post-processed data.”

And on p. 17, 18 we write (for protein data processing):

“To derive a high-confidence dataset, we removed all genes with one or more missing data points, resulting in 1,820 genes for further processing. This dataset was normalized by the sum of all LFQ intensities, where the sum excluded the top 5% intensities in each sample. Removing the top 5% most intense proteins from the summation can prevent extremely abundant proteins (and potential outliers) from dominating the normalizing factor. The data was then log transformed (natural base) and the jackknife procedure was applied to remove outliers (i.e. spikes) in the data, similar to what was described above for the transcriptomics data. The TRV threshold was 2 upon examination of the histograms of all TRVs (not shown). Subsequently, we applied weighted local regression (Lowess) to further smooth the time course data. A total of 1,237 genes were left in final dataset with complete and post-processed, high-confidence mRNA and protein annotations for two replicates across eight time points.”

Proteomics samples were adjusted to the same protein concentration per sample, as stated in the Methods section, p. 16, 17.

Finally, we added a figure to the Appendix Notes in which we show the expression levels of housekeeping genes (Figure S12), i.e. actins, tubulins, and histones. The expression of these proteins is comparatively constant across the time course. We mention the additional figure on p. 8 of the main text.

3. In the method section, the authors should add information on MS measurements and data analysis. Particularly, the fragmentation mode (CID/HCD, ...), normalized collision energy, dynamic exclusion, etc. It would be also beneficial to add MaxQuant output tables as a supplementary data so the readers could easily access all search parameters (parameters.txt) and results (summary.txt, evidence.txt and proteinGroups.txt). Many questions related to data analysis could be answered by providing Perseus session file (with .sps extension) as it contains all data analysis steps.

We amended the Methods description accordingly, to write on p. 17:

“LC-MS/MS Analysis

Peptides were separated by reverse phase nanoflow high-performance liquid chromatography (nano-HPLC) and quantified on an LTQ-Orbitrap Velos mass spectrometer (Thermo Scientific). Data-dependent analysis was performed at a resolution of 60,000 and with the top 20 most intense ions selected from each MS full scan, with dynamic exclusion set to 90 s if m/z acquisition was repeated within a 45 s interval. In each scan cycle, the top 20 fragmentation spectra were acquired in the collision-induced dissociation mode. For peptide separation, we used an Agilent ZORBAX 300SB-C18 reverse phase column (150 mm x 75 µm inner diameter) and a 240 min gradient of 2 to 90% acetonitrile and 0.1% formic acid. We conducted the experiment twice (biological replicates) and collected four technical replicates for each sample (repeat mass spectrometry measurements). The data for technical replicates were combined during the computational analysis (see below), while the biological replicates were kept separately.

Proteomics data processing and quality control

Raw data were processed using the MaxQuant software (1.3.0.3) (Cox and Mann, 2008) and peak lists were searched with Andromeda (Cox et al., 2011) against a database containing the translation of all predicted proteins listed in UniProt (2012), and a list of commonly observed contaminants supplied by MaxQuant. Protein identification was performed using 20 ppm tolerance at the MS level (FT mass

analyzer) and 0.5 Da at the MS/MS level (Ion Trap analyzer), with a posterior global FDR of 1% based on the reverse sequence of the human FASTA file. Up to two missed trypsin cleavages were allowed, and oxidation of methionine and N-terminal acetylation were searched as variable post-translational modification; cysteine carbamidomethylation as fixed. The minimal required peptide length was set to seven amino acids and both protein. The minimum number of peptide pairs used for quantitation was set to one. We used MaxQuant to combine the cytosolic and pellet samples as different fractions in the same experiment, after testing different options (Appendix Figure S8).”

The output text files of the MaxQuant search, containing information parameter settings, experimental setup, a protein LFQ intensities etc have been part of the PRIDE database submission of the data. However, we also added Appendix Datafile 3 which contains *all* MaxQuant output text files. As much of the data normalization and the PECA analysis was performed with tools other than Perseus, we could not provide a .sps file. All program files for the PECA tool are available as stated in the original publication (*J Prot Research 2014*).

4. On p. 5 (and method section p. 14), the authors should provide additional information on how they compare kinetic profiles of individual gene/protein from transcriptomics/proteomics datasets and how clusters were selected to identify specific trends (e.g. concordant or discordant clusters between omics datasets).

We revised the discussion of discordant (and concordant) gene expression. As the discussion of discordance is a minor part of the results, we placed most of its description into the Appendix (Figure S16).

Results, p. 9 “Next, we asked if mRNA- and protein-level regulation occurred in a concordant fashion, i.e. in the same direction, or discordantly, i.e. working in opposing directions. ... Table 1 already indicates that discordant regulation is comparatively rare: only few genes are listed in the lower left and upper right corners of the tables (75 genes in total). Alternative ways to identify discordant regulation confirmed this result, i.e. via filtering for negative correlation between PECA’s mRNA and protein rate ratios across both replicates (Appendix Dataset 1, Appendix Figure S16).”

With the description of Appendix Figure S16 “First, we aimed to assess the extent to which discordant regulation occurs, i.e. in how many genes mRNA- and protein-level regulation occur in opposite directions. To do so, we calculated the Pearson correlation was in each replicate (A., B.) between the time course RNA- and protein-level rate ratios for each gene. Genes with (i) significant RNA- and protein-level CPS (FDR<0.05) at any time point and (ii) rate ratio correlation below -0.5 were considered to be subject to discordant regulation, i.e. counteracting regulation at the mRNA and protein level. Fifty-five genes adhered to this definition in both replicates and are listed in Appendix Dataset 1. In the heat map, the rate ratios were normalized against the first time period in both RNA and protein level analyses. C. In a modified analysis, we attempted to extract genes that were candidates for ‘buffering’, i.e. in which discordant mRNA- and protein-level regulation results in constant protein concentrations despite changes in mRNA expression. Again, we extracted genes whose PECA analysis resulted in at least one significant change (FDR<0.05) at the mRNA- and one at the protein-level, in opposing directions. We filtered this set further for protein concentration changes of less than 1.5-fold (normalized, relative log-ratios). This procedure resulted in three genes shown in this figure.”

5. Figures S4 and S7 show good correlation between individual measurements. However, the comparison of dynamic trends (e.g. effect of "buffering" on figure S11) is less consistent. Why the overlap between proteins with "buffering effect" (see Dataset1 excel file, tab "Buffering") is so small (e.g. 52 genes in R1, 59 genes in R2 and only 10 genes common to both R1 and R2)? Since

PECA analyzes temporal profiles of mRNA/protein it would be important to see a good correlation of temporal patterns between replicates, a point that has not been addressed in the manuscript. The lack of consistency between replicates is also evident from Figures 2 and S9. The authors should display protein/mRNA expression temporal profiles for all "buffering" genes to compare profiles between replicates. The only cases presented are for splice variants in QARS (Fig S14), where trends are similar though the shapes of the temporal profiles are different. Since PECA analysis is using the shape of the temporal profile rather than just a up/down trend, these do not appear to be representative examples of good correlation between replicates.

To address the reviewer's comments, we expanded the discussion on reproducibility between the two biological replicates in both the main text (Results, p. 7) and the Appendix (Figure S13C).

The Appendix Notes Figure 13C shows the frequency distribution between the replicate time series measurements for RNA and protein, respectively, i.e. matching for each gene the eight mRNA concentrations from replicate 1 with those from replicate 2, and the same for protein concentration measurements. The distributions are shifted towards high values suggesting high reproducibility, but some genes have low correlation. There are several reasons for low correlation: i) the peak expression change occurs at 2 hours in one replicate, but at 8 hours in the other (shifted); ii) the overall expression change is very small, i.e. <1.5-fold, but with different profiles across the replicates (e.g. HSC70, see below); or iii) the biological replicates truly disagree for unknown reasons.

(Note that Appendix Figures S6 and S10 also show Figures that reproducibility and consistency between replicates is generally high, with an average $R > 0.97$ and $R > 0.94$ for the RNA and proteomics data, respectively (with the exception of time=30 hours with $R = 0.88$).)

To enable the reader's assessment of reproducibility between replicates, but without overcrowding the figures and distracting from the main results, we added a new measure to the descriptions in the main text. We call this measure Replicate Consistency Measure (RCM) and it displays the Pearson correlation coefficients between replicate RNA and protein profiles for a gene. The RCM values are now mentioned for all example genes discussed in the text.

In the light of Reviewer #2's comments, we edited and shortened the discussion of buffering and discordant gene expression, which have been discussed in previous answers (see above).

The profiles for the analysis of splice variants are much noisier as the analysis was conducted on a different dataset. For the main analyses, we used MaxQuant's *proteinGroups.txt* output file which sums over all peptides contributing to a protein group. This dataset was then normalized as described in the main text and above. In comparison, the analyses of splice variants was conducted using the *peptides.txt* file to obtain information on individual peptides that mapped to different splice variants. Appendix Figure S17 shows this raw, un-normalized data. We added this note to the legend of Figure S17 to clarify this point.

We edited the main text to show RCM measures to all example genes, and as shown below.

Results, p. 7 "A variety of tests confirmed the quality of the proteomics data, e.g. western blots of selected proteins and analysis of housekeeping genes, and its reproducibility across the two biological replicates (Appendix Figures S11-S13). We ... chose a high-confidence dataset of 1,237 proteins for further analysis with complete time series measurements across both replicates. ... Protein concentrations ... reproducibility was high ($R > 0.94$ for seven of the eight time points, Appendix Figure S10); the correlation with the corresponding mRNA concentrations was consistent across samples (Appendix Figure S13). Heatmaps of the integrated and clustered

mRNA and matching protein expression values show that overall expression changes were similar between the two biological replicates (Figure 2, S5, S9, S14), but some discrepancies exist. In some cases, peak expression changes occurred at 2 hours in one replicate and at 8 hours in the other. To describe experimental reproducibility, we calculated a Replicate Consistency Measure (RCM) that lists the Pearson correlation coefficient between replicate time series measurements of normalized, log-transformed RNA and protein concentrations. At a total of eight datapoints, a Pearson correlation coefficient >0.7 corresponds to a P-value=0.05. For example, for GRP78 the RCM is 0.87/0.97, suggesting high reproducibility between the two biological replicates. Appendix Figure S13 displays the frequency distributions of all RCM values which show a bias towards high values.”

6. On p. 7, p. 9 and several other places throughout the text. The term "switch-like manner" could be misleading considering time resolution of the experiment, if the authors observe change only between 2 time points this does not necessarily imply a switch-like manner. Using higher temporal resolution might reveal slow steady-state changes instead.

The revised manuscript compares the terms ‘spike-like’ and ‘switch-like’ to ‘transient’ and ‘permanent’ changes in the regulatory profiles to clarify these concepts. We added a note to the Discussion on p. 12 “We note that without higher temporal resolution it is impossible if such a switch is indeed very rapid or more continuous over a period of time.”.

7. Dataset1 table contains individual proteins in separate rows. In the proteomics experiments several peptides can be shared among the homologous proteins, and MaxQuant reports them as a protein group. How was this information reported in the table?

We added a note to p. 16/17 “As is expected from complex proteomes such as that from mammalian cells, peptides can be shared between homologous proteins or splice variants, leading to ‘protein groups’. The protein group structure is shown in MaxQuant’s *proteinGroups.txt* file in the Appendix Dataset 3; for clarity, we refer to the first and main protein from each group throughout the text.”

8. On p. 12 "We devised a jackknife procedure to remove aberrant expression measurements". I'm not sure if this procedure is applicable for the selected time frame (e.g. time points: 0, 0.5, 1, 2, 8, 16, 24 and 30 h). I would understand if the measurements were performed every 5 min, and a sudden spike is probably a technical artifact. Under the present condition, an increase in abundance between 8 to 16 h followed by a decrease to original level at 24h would be interpreted as a spike, a situation that might not be realistic for these time points.

The removal of outlier measurements (‘spikes’) is not only based on a single expression profile, but on variation across the biological replicates and across *all* genes. We devised these strict filtering rules to derive a high-confidence dataset with reliable information that is reproducible between the replicates.

We extended the description of the procedure to clarify these points, p. 15 “We devised a jackknife procedure to remove aberrant expression measurements. In the procedure, one data point was removed at a time and the total range of variation (TRV), defined as the difference between maximum and minimum, for the remaining dataset was recorded. After following this procedure for all data points, the ratio of the median TRV to the minimum TRV was calculated and used as a measure of ‘spikiness’ (or noisiness) of the data. The larger the TRV, the noisier the gene. After examining the histogram of the TRV values across all genes (*not shown*), the threshold for tolerance level was set to 3 from the histogram of TRV values across *all* genes, and genes below the threshold were retained. These strict filtering rules enable construction of a high-confidence dataset albeit possibly removing true signal.”

9. On p. 13 "Finally, locally weighted scatter plot smoothing (lowess) was applied to further smooth the filtered data for robust estimation of kinetic parameters in the PECA model." More details are required. With lowess one can smooth any profile even if the actual data look terrible, what were the parameters used for the smoothing and which level of RSD was considered as acceptable?

We amended the description of the smoothing as shown below and also include an example plot (for GRP78) with the original data in dotted lines, and the post-processed data in continuous lines. All such plots are shown in Appendix Dataset 4.

Methods, p. 15 "Finally, locally weighted scatter plot smoothing (Lowess) was applied to further smooth the filtered data for robust estimation of kinetic parameters in the PECA model. To do so, we used the *lowess* function in R, a standard implementation of local weighted scatter plot smoothing, with the default parameter settings. We manually inspected the extend of smoothing: the large majority of time course profiles changed very little, except for those with zig-zag patterns between time points. Appendix Dataset 4 shows the original and post-processed data."

10. Proteomics experiments (p.13). Was there a rationale for separating nuclear and cytosolic fractions since no comparison was performed between fractions?

We separated the two subcellular fractions to increase proteomic coverage. We added Appendix Figure S8 to compare the performance between fractions and explain their computational integration.

Minor comments.

1. Figure S10, figure legends (Protein-1/Protein-2) are not explained in the text.

This has been corrected in the figure (now Figure S15) and in Figure S20 – "RNA-1, Protein-1 and RNA-2, Protein-2 refer to replicate 1 and 2, respectively."

2. Description in the file Dataset 1 needs more details (e.g., are numbers in the column "Protein replicate 1" represent intensity, what do the column "Protein rel rep1" correspond to?)

We amended these descriptions.

3. Check text for consistency (e.g. p. 13 "Each sample was by with three to four" something is missing here).

We corrected this description, p. 16 to "We conducted the experiment twice (biological replicates) and collected four technical replicates for each sample (repeat mass spectrometry measurements). The data for technical replicates were combined during the computational analysis (see below), while the biological replicates were kept separately."

2nd Editorial Decision

30 November 2015

Thank you again for submitting your work to Molecular Systems Biology. We have now heard back from the two referees who agreed to evaluate your manuscript. As you will see below, the referees think that their main concerns have been satisfactorily addressed. However, referee #2 expresses concerns regarding the statement on "concordant mRNA-protein regulation", which we would ask you to address in a revision of the manuscript.

REFeree COMMENTS

Reviewer #2:

The authors have addressed most of my concerns raised in the review of the initial submission of the paper and have provided a substantially improved manuscript. Especially the re-analysis of the data set from Jovanovic et al 2015 significantly helped to relate their findings to previous knowledge and therefore to generalize the conclusions..

However I do have an additional concern on their new claim about "concordant mRNA-protein regulations" (Page 9, Para 2 in Result & Page 12, Para 3 in Discussion). Here the authors argue that, according to Table 1, "only few genes are listed in the lower left and upper right" (75 genes in total), which indicates the mRNA-protein discordant regulation is rare. However, if one counts the gene numbers listed in the "upper left" and "lower right" (which should then indicate the "concordant regulations" according to the author)-- there are also only 95 genes. Additionally, based on their Appendix Figure S16A/ B, it seems to this reviewer that, the low number of "discordant genes" was largely due to the arbitrary cutoff of -0.5 for the rate ratio correlation coefficient (R). In fact, according to their histogram distribution of correlations (Figure S16 A/B), those genes with $R < -0.5$ seem to be more numerous than those with $R > 0.5$, which could indicate a different conclusion. Finally I found the claim of "discordant mRNA-protein regulations are rare" weakens the main argument that mRNA and protein level regulations have significantly different dynamics (spike-like vs. switch- like). Therefore I suggest revising this by tuning down or removing this claim.

Reviewer #3:

The authors have addressed most of my queries in a satisfactory fashion, though the issue of reproducibility across the two biological replicates still remains a concern.

2nd Revision - authors' response

04 December 2015

Detailed response

Our replies are printed in bold font.

Reviewer #2:

The authors have addressed most of my concerns raised in the review of the initial submission of the

paper and have provided a substantially improved manuscript. Especially the re-analysis of the data set from Jovanovic et al 2015 significantly helped to relate their findings to previous knowledge and therefore to generalize the conclusions.

However I do have an additional concern on their new claim about "concordant mRNA-protein regulations" (Page 9, Para 2 in Result & Page 12, Para 3 in Discussion). Here the authors argue that, according to Table 1, "only few genes are listed in the lower left and upper right" (75 genes in total), which indicates the mRNA-protein discordant regulation is rare. However, if one counts the gene numbers listed in the "upper left" and "lower right" (which should then indicate the "concordant regulations" according to the author)-- there are also only 95 genes. Additionally, based on their Appendix Figure S16A/ B, it seems to this reviewer that, the low number of "discordant genes" was largely due to the arbitrary cutoff of -0.5 for the rate ratio correlation coefficient (R). In fact, according to their histogram distribution of correlations (Figure S16 A/B), those genes with $R < -0.5$ seem to be more numerous than those with $R > 0.5$, which could indicate a different conclusion. Finally I found the claim of "discordant mRNA-protein regulations are rare" weakens the main argument that mRNA and protein level regulations have significantly different dynamics (spike-like vs. switch-like). Therefore I suggest revising this by tuning down or removing this claim.

We have removed all sentences suggesting that there are more proteins subject to concordant regulation than discordant regulation between the two levels during the same phase of the experiment. Instead, we now state that most of the regulatory changes at the two molecular levels are independent at particular time points, but often concordant in their final outcome.

p. 7/8: "Table 1 shows the numbers of significant regulatory events for one of the replicates, grouped according to phase, level, and direction of the regulation. [...] The numbers are symmetrically distributed across the table, confirming the observation from Figure 4E, F that mRNA- and protein-level regulation contribute equally to the overall gene expression changes in this experiment, affecting similar numbers of genes. As Table 1 shows, if a gene was significantly regulated during a specific phase of the response, this regulation typically occurred at either the mRNA- or the protein-level, but not at both at the same time; the numbers of genes in each of the square's corners are smaller than those in the middle rows or columns. However, some genes showed mRNA- and protein-level regulation moving in the same direction during the same phase, others showed movement in opposite directions."

p. 11: "While most concentration changes were seemingly concordant between mRNA and protein in terms of the outcome at the end of the experiment, many regulatory (rate ratio) changes, in particular the most pronounced ones, were independent between the mRNA and protein level with respect to their timing or direction. We even observed a small number of cases in which transcript and protein level regulation acted in opposite directions. One example is the ER stress related chaperone GRP78, whose mRNA concentration was in decline at 16 hours, while protein concentrations still increased (Figure 3). We find that discordance at a specific time point was often resolved by a simple delay in the response: the changes at the RNA level are initially counteracted at the protein level, but later supported by concordant action. Overall, true 'buffering' appears to be the exception within the set of proteins we surveyed, and most regulatory events at the mRNA and protein level are coordinated, albeit with different timing, to achieve a new proteomic state."

Reviewer #3:

The authors have addressed most of my queries in a satisfactory fashion, though the issue of reproducibility across the two biological replicates still remains a concern.

The majority of the genes in our dataset show similar changes across both mRNA and protein expression data; but for a small fraction there are differences between the

replicates. For this reason, the results we report rely on findings that were observed in both replicates. We added a note to the Discussion, p. 11: “Further, we focus on results that are consistent across the two biological replicates, and observed in a high-confidence dataset without missing data.”